# A Large-scale Validation of Snowpack Simulations in Support of Avalanche Forecasting Focusing on Critical Layers

Florian Herla[1], Pascal Haegeli[1], Simon Horton[2], and Patrick Mair[3]

[1]Simon Fraser University, Burnaby, BC, Canada
[2]Avalanche Canada, Revelstoke, BC, Canada
[3]Harvard University, Cambridge, MA, USA

**Correspondence:** Florian Herla (fherla@sfu.ca)

**Abstract.** Avalanche warning services increasingly employ snow stratigraphy simulations to improve their current understanding of critical avalanche layers, a key ingredient of dry slab avalanche hazard. However, a lack of large-scale validation studies has limited the operational value of these simulations for regional avalanche forecasting. To address this knowledge gap, we present methods for meaningful comparisons between regional assessments of avalanche forecasters and distributed snowpack simulations. We applied these methods to operational data sets of ten winter seasons and three forecast regions with different snow climate characteristics in western Canada to quantify the Canadian weather and snowpack model chain's ability to represent persistent critical avalanche layers.

Using a recently developed statistical instability model as well as traditional process-based indices, we found that the overall probability of detecting a known critical layer can reach 75 % when accepting a probability of 40 % that any simulated layer is actually of operational concern in reality (i.e., precision) as well as a false alarm rate of 30 %. Peirce skill scores and F1 scores cap at approximately 50 %. Faceted layers were captured well but also caused most false alarms (probability of detection up to 90 %, precision between 20–40 %, false alarm rate up to 30 %), whereas surface hoar layers, though less common, were mostly of operational concern when modeled (probability of detection up to 80 %, precision between 80–100 %, false alarm rate up to 5 %). Our results also show strong patterns related to forecast regions and elevation bands and reveal more subtle trends with conditional inference trees. Explorations into daily comparisons of layer characteristics generally indicate high variability between simulations and forecaster assessments with correlations rarely exceeding 50 %. We discuss in depth how the presented results can be interpreted in light of the validation data set, which inevitably contains human biases and inconsistencies.

Overall, the simulations provide a valuable starting point for targeted field observations as well as a rich complementary information source that can help alert forecasters about the existence of critical layers and their instability. However, the existing model chain does not seem sufficiently reliable to generate assessments purely based on simulations. We conclude by presenting our vision of a real-time validation suite that can help forecasters develop a better understanding of the simulations' strengths and weaknesses by continuously comparing assessments and simulations.

# 1 Introduction

Understanding snow avalanche hazard conditions requires information about the layered snow stratigraphy (LaChapelle, 1980; McClung, 2002b). Traditionally this information is gathered with manual snow pit observations at targeted point locations (McClung, 2002a; Campbell et al., 2016). Avalanche forecasters then combine this information with their prior mental model of the snow conditions and information about the weather as well as avalanche occurrences to evaluate the conditions at the scale of the forecast application (e.g., regional public forecast) (LaChapelle, 1980). Of particular interest are the distribution

and sensitivity of persistent weak layers and crusts (Statham et al., 2018a) that were buried over the course of the winter season and persist across the region, since these layers can cause enduring avalanche problems with multiple recurrent avalanche cycles. While assessing the current snowpack conditions (i.e., nowcasting) for entire regions based on limited data availability is already an exercise that demands considerable expertise, forecasting for snowpack changes several days ahead requires considerable subjective judgment that can be associated with substantial uncertainty (McClung, 2002a; Statham et al., 2018b).

This paper examines the effectiveness of the Canadian operational weather and snowpack model chain to identify critical avalanche layers that are essential for regional-scale avalanche forecasting.

  Weather and snowpack model chains represent an additional, independent data source for detailed snow stratigraphy information with high spatiotemporal availability (Morin et al., 2020). Snowpack simulations can therefore help fill data sparsity gaps occurring during the early winter season, in remote regions (Storm, 2012; Storm and Helgeson, 2014), or they can com-

plement assumptions on the evolution of the sensitivity to triggering avalanches on known critical layers (Reuter et al., 2021; Mayer et al., 2022). However, the simulations inherently accumulate errors from the weather inputs—precipitation being the main source of uncertainty for snowpack structure and instability (Raleigh et al., 2015; Richter et al., 2020)—and from deficiencies in the snow model formulations (Lafaysse et al., 2017). Thus, avalanche forecasters are concerned about the validity of the simulations and only hesitantly integrate these novel information sources in their operational assessment processes (Morin

et al., 2020).

  Over the last 20 years, extensive research has been conducted to improve the capabilities of snowpack models and explore their application for avalanche forecasting. Many studies that validated simulated snow stratigraphy applied a process-based approach (e.g., Brun et al., 1992; Fierz, 1998; Durand et al., 1999; Lehning et al., 2001, 2002b; Fierz and Lehning, 2004; Bellaire and Jamieson, 2013; Reuter and Bellaire, 2018; Richter et al., 2019; Calonne et al., 2020; Viallon-Galinier et al., 2020).

These process-based studies tested the capabilities of the models to represent specific physical processes, mostly using high-quality validation data sets at the point scale. On the regional scale, much effort has gone into validating bulk snow properties such as snow depth or snow–water equivalent (Vionnet et al., 2012; Lafaysse et al., 2013; Quéno et al., 2016; Vionnet et al., 2019; Morin et al., 2020; Horton and Haegeli, 2022). Several studies evaluated the model capabilities to simulate snowpack instability based on regional observations to support its application in avalanche forecasting (Schweizer et al., 2006; Schirmer

et al., 2010; Vernay et al., 2015; Bellaire et al., 2017; Mayer et al., 2022).

  Despite this large body of snowpack validation studies, the operational needs of (Canadian) avalanche forecasters have not been satisfied yet. While process-based validations at individual point locations based on high quality data are crucial for

model development and improvement, these validation results do not provide sufficiently tangible and relevant guidance for forecasters who forecast for different locations or regions. In addition, these validation results are not necessarily representative of the real skill of operational simulations which might rely on different data sources or model configurations. Regional-scale validation studies of simulated snowfall further contribute essential information to the valuation of snowpack simulations for avalanche forecasting, particularly with respect to snow surface avalanche problems (e.g., storm snow problems) and characteristics of the slab, which is primarily influenced by precipitation (Richter et al., 2020). Nevertheless, the existing research does not paint a comprehensive picture yet: to our knowledge no large-scale study exists that created a specific link between simulated layers and known critical avalanche layers. Forecasters therefore only have a limited understanding of how to interpret individual critical layers in the simulations. To address these validity concerns and increase the operational adoption of snowpack simulations in avalanche forecasting, forecasters need an application-specific validation approach at the regional scale that evaluates the detailed hazardous layering of the snowpack, most optimally in an operational, real-time format. The only study that followed a similar approach is Horton and Jamieson (2016), who uncovered substantial uncertainties in the prediction of hazardous surface hoar formation and its post-burial evolution during two winter seasons. All of these observations highlight the importance of an updated validation study and future real-time validation tools.

The objective of this study is to evaluate the performance of operational snowpack simulations to represent persistent weak layers and crusts in support of operational avalanche forecasting at the regional scale. Based on forecaster assessments in ten winter seasons and three mountain ranges, we characterize the overall model performance with the probability that hazardous layers are captured by the simulations (i.e., probability of detection) and the probability that modeled critical layers are actually of concern in reality (i.e., model precision). To make our results most informative for forecasters, we explore patterns in the presence or absence of persistent weak layers and crusts that can be explained with attributes of the layers themselves, and we present the results in tangible ways that help forecasters interpret daily model scenarios. We also quantify the degree of agreement between simulated and reported layer instability using indicators of the variation and timing of instability. By relying on operationally available data from human avalanche hazard assessments, the methods developed in this study can be used to design future operational validation suites. Overall, our study quantifies the capabilities of an operational weather and snowpack model chain to represent critical avalanche layers that are prone to cause dry slab avalanches, which contributes to making snowpack simulations more transparent and applicable for operational applications.

We continue this paper by introducing the used data sets in Sect. 2 before explaining our methodology in detail in Sect. 3. Section 4 presents the results from their highest level to the most detail. We direct our discussion in Sect. 5 first to avalanche practitioners and share the insights gained through this study for operational applications as well as share our vision for the future development of operational tools before discussing insights relevant for the snowpack modeling community.

## 2  Data

The data sets used in this study consist of snowpack simulations and operational avalanche hazard assessments from avalanche forecasters in western Canada over ten winter seasons (2013/14–2021/22). This section provides the necessary background

| | TOTAL | alpine (ALP) | treeline (TL) | below treeline (BTL) |
|---|---|---|---|---|
| Sea-to-sky (S2S) | 476 | 48 (10%) | 54 (11%) | 374 (79%) |
| Glacier National Park (GNP) | 233 | 30 (30%) | 102 (44%) | 101 (43%) |
| Banff–Yoho–Kooteney (BYK) | 295 | 7 (2%) | 163 (55%) | 125 (42%) |

**Table 1.** Number and percentage of model grid points in each region and elevation band.

information on the study area (Sect. 2.1), the snowpack simulations employed for this study (Sect. 2.2), as well as the human hazard assessments used as validation data set (Sect. 2.3). In the manuscript, each winter season is defined to span from December of the previous year through March of the stated year (e.g., winter season 2021/22 will be referred to as 2022).

## 2.1 Study area

The study focuses on three data-rich public avalanche forecast regions in British Columbia and Alberta, Canada, each one characterized by a different snow climate: Whistler in the Sea-to-Sky region (S2S) in the maritime Coast Mountains, Glacier National Park (GNP) in the transitional Columbia Mountains, and Banff-Yoho-Kooteney National Park (BYK) in the continental Rocky Mountains (Shandro and Haegeli, 2018). A comprehensive table of acronyms (Table A1) and table of variables (Table A2) can be found in Appendix B.

While the human assessment data set is inherently compiled for these distinct forecast regions, we had to select model grid points to represent these regions in the simulations. We used all grid points within the region boundaries that were within 10 km of accessible roads for GNP and BYK, and within 20 km of Highway 99 for S2S. Since Canadian National Parks prohibit motorized terrain access, such as via sleds, we assumed the hazard assessments to be highly biased towards these parts of the regions. Overall, we selected 1004 grid points (Fig. 1) covering area of 6275 km$^2$.

We classified each grid point into an elevation band class, 'alpine' (ALP), 'treeline' (TL), and 'below treeline' (BTL) to match the terrain classification in the human assessments. To create the best possible match in elevation between assessments and simulations, we used forecaster consensus of TL elevation for the classification: 1600–1800 m asl in S2S, 1800–2100 m asl in GNP, and 2000–2400 m asl in BYK. Table 1 describes the distribution of model grid points across the forecast regions and elevation bands. Due to the configuration of our snowpack simulations (flat field, no wind transport, see next section, 2.2),
we do not discuss slope aspects or prominent wind directions across our study areas.

## 2.2 Snowpack simulations

The snowpack simulations used in this study mirror the setup of the operational simulations that are provided to Canadian avalanche forecasters. The raw output of the numerical weather prediction model HRDPS (High Resolution Deterministic Prediction System, Milbrandt et al., 2016) is fed into the detailed snow cover model SNOWPACK (Bartelt et al., 2002;
Lehning et al., 2002a, b) (v3.4). HRDPS is run operationally by the Meteorological Service of Canada on a 2.5 km horizontal

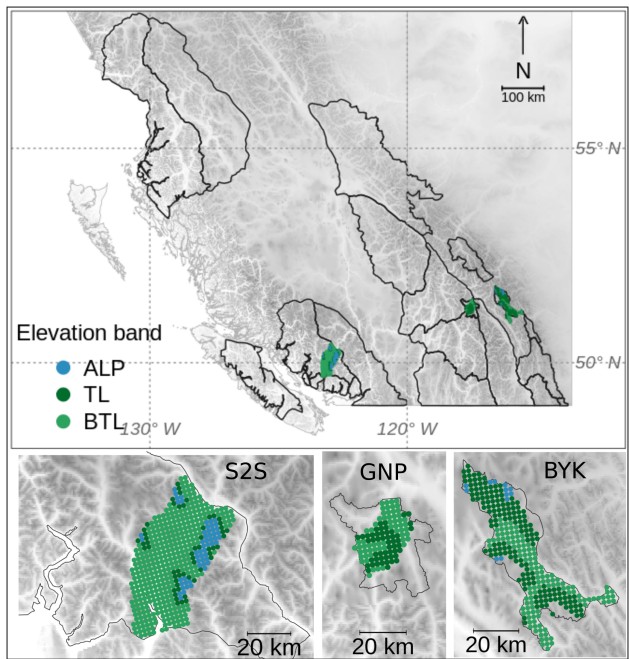

**Figure 1.** The public avalanche forecast regions of western Canada. The model grid points selected for this study are visualized by their elevation band (alpine [ALP], treeline [TL], below treeline [BTL]) and lie within the forecast regions Sea-to-Sky (S2S), Glacier National Park (GNP), and Banff-Yoho-Kootenay National Park (BYK). The grid has a 2.5 km spacing.

grid. SNOWPACK is run with weather data lead times of 6–12 h, using air temperature and relative humidity at 2 m above ground, wind speed at 10 m above ground, incoming shortwave and longwave radiation fluxes at the surface, and accumulated precipitation. The simulations were initialized in September without any snow on the ground. We used the atmospheric stability scheme by Michlmayr et al. (2008) implemented in SNOWPACK, and turned off any snow redistribution by wind. Lastly, we
turned off SNOWPACK's layer aggregation feature of merging similar and adjacent layers to preserve exact knowledge of the formation dates of individual layers. The need for this decision will become apparent in Sect. 3.1.2 and 3.1.3, where we explain our approach of grouping and matching layers based on date considerations. All simulated profiles were valid between 4–5 PM local time representing flat field conditions.

To visualize the grain types of snow layers, this manuscript uses the hazard-focused color coding suggested by Horton et al.
(2020a). We abbreviate grain types with the following acronyms: precipitation particles (PP), decomposing and fragmented precipitation particles (DF), surface hoar (SH), depth hoar (DH), faceted crystals (FC), rounding faceted particles (FCxr), rounded grains (RG), rain crust (IFrc), sun crust (IFsc), or temperature/melt–freeze crust (MFcr), and melt forms (MF). All grains types are defined in the International Classification for Seasonal Snow on the Ground (Fierz et al., 2009). Figure 2 visualizes the output of the snowpack model at an individual grid point. The time series view illustrates how the seasonal

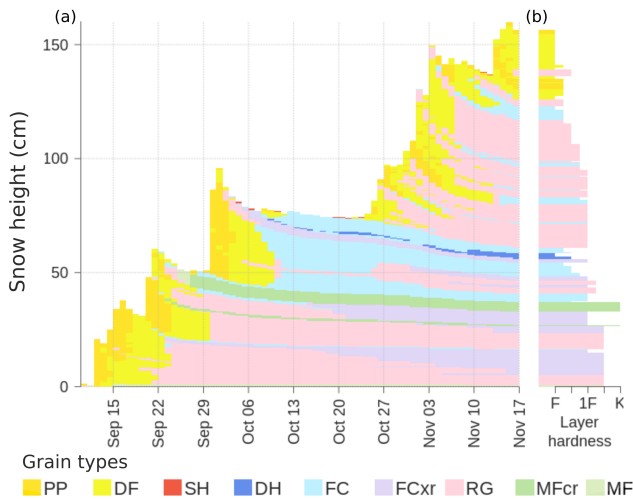

**Figure 2.** A demonstration of a simulated snow profile at an individual model grid point. (a) A time series view that illustrates how the snowpack builds up over the winter season in a layered structure. (b) Profile view at a single time step highlighting layer hardness. Colors refer to snow grain types; acronyms for these grain types are defined in the text and Table A1.

snowpack builds up from the beginning of the winter season in a layered structure (Fig. 2a), while each layer is associated with structure-mechanical properties at every time step, such as layer hardness (Fig. 2b).

## 2.3 Avalanche hazard assessments

Avalanche hazard assessments compiled for this study were issued by public avalanche forecasters every day of the winter season. The assessments represent forecasters' best knowledge of the current conditions (i.e., nowcasts) and were issued in the afternoon for one elevation band (ALP, TL, BTL) in one forecast region. Applying the Conceptual Model of Avalanche Hazard (Statham et al., 2018a), forecasters partitioned the avalanche hazard into different avalanche problems and characterized each problem by its type, location, likelihood of avalanches (resulting from spatial distribution and sensitivity to triggering), and destructive avalanche size (Fig. 3a).

In addition to avalanche problems, the hazard assessments also contain records about persistent weak layers and crusts (Fig. 3b). Forecasters typically track the evolution of these layers and associate them with the relevant avalanche problems at their times of concern. To facilitate the tracking across space and over time, avalanche forecasters name these layers with date tags and their grain type(s), e.g. "Jan 17th surface hoar layer". Reported date tags mostly represent the beginning of snowfall periods that bury layers that were exposed to the snow surface before the snowfall and are therefore likely to contain weak grain types. Sometimes the date tags can also represent rain events that form a crust at the snow surface. While the date tags provide general markers to simplify communication, the actual formation or burial dates can vary within a forecast region due to differences in local weather and its timing. When a tracked layer is associated with an avalanche problem at a given day, the forecaster expects the problem to be governed by that layer.

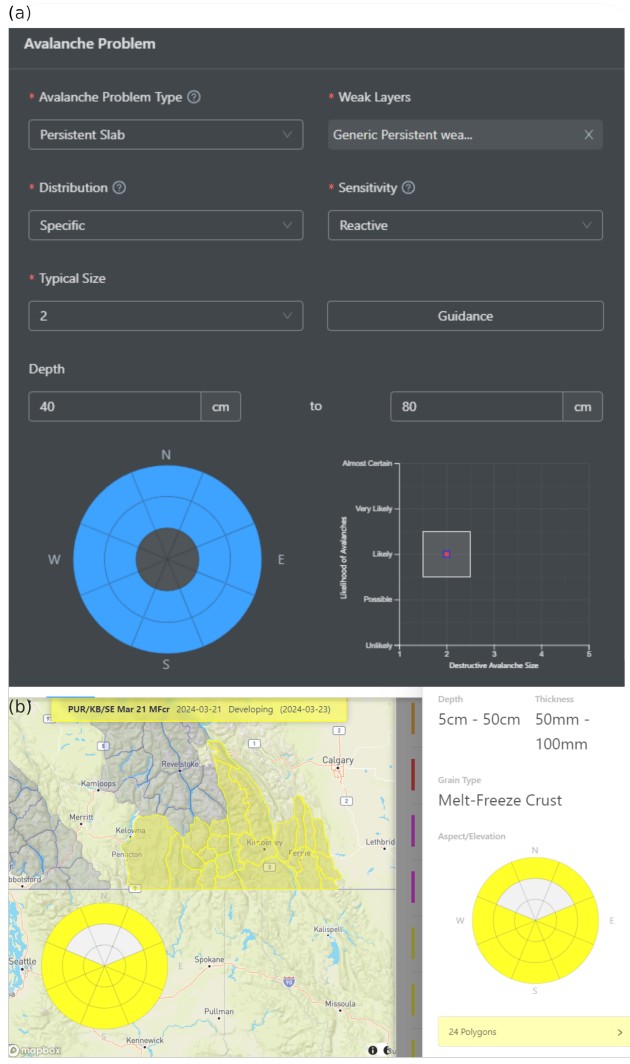

**Figure 3.** Screenshots of the dashboard used by avalanche forecasters for recording hazard assessments. (a) The entry tool for avalanche problem characteristics, which allows linking specific avalanche problems to the weak layers documented in (b) the weak layer tracking tool. Both tools integrate assessments with forecast subregions selected through an interactive map.

The grain types of the tracked layers were either reported as one specific grain type or a mix of grain types, such as SH, FC, DH, IFrc, IFsc, or more generically MFcr, where each layer is usually primarily a persistent weak layer (SH, DH, FC) or a crust layer (IF, MFcr).

## 3 Methodology

Our validation approach consists of three main components. First, we assessed the general performance of the simulations with regard to modeling persistent weak layers and crusts (Sect. 3.1) by examining two practical questions: a) What is the probability that a layer of operational concern is captured by the simulations? and b) How likely is it that a modeled critical layer also exists in reality? Second, we explored whether there are distinguishable patterns between layers of concern that were captured well by the model and those that were not (Sect. 3.2). Finally, we examined the agreement between simulated and reported layers in more detail taking into account the timing and variation of layer instability (Sect. 3.3).

### 3.1 Assessing the general simulation performance

We used a confusion matrix approach to quantify the performance of the simulations to represent persistent weak layers and crusts. A confusion matrix, or a 2x2 contingency table, presents the joint frequency distributions of binary forecasts and observations (Wilks, 2019, p. 374f) as the foundation for calculating various performance attributes. In our case, the binary events were the presence or absence of persistent weak layers and crusts at some point during the season in the human hazard assessments and snowpack simulations. To identify present and absent layers in both data sets and properly fill the four cells of the confusion matrix, we had to perform several pre-processing steps. This involved a) identifying the relevant layers in both data sets, b) matching the corresponding layers, and c) translating the spatially distributed simulation information into a binary variable representing presence or absence at the regional scale. We describe each of these steps in detail in the following sections, and Figure 4 provides a concise summary of the process.

#### 3.1.1 Pre-processing human data set

We identified relevant layers in our human hazard assessment data set by filtering all layers that were tracked by avalanche forecasters to only those that were associated with either storm slab, persistent slab or deep persistent slab avalanche problems at least once during a season. Since these layers are a key ingredient to these avalanche problems, we refer to them as *layers of concern* throughout the manuscript. In addition, we will refer to persistent and deep persistent avalanche problems simply as persistent problems.

To account for the various degrees of concern and level of confidence in the human assessments, we manually reviewed all forecaster comments associated with the identified layers of concern and summarized them in a qualitative data quality rating. The forecaster comments often include information about observed avalanches, associated triggers, other instability observations, the absence of observed instability, or mere assumptions. Combining the qualitative comments with the number of avalanche problem days and the assessed likelihood of triggering avalanches on these layers, we assigned each layer of concern to one of four data quality classes. Class 1 (Excellent) contains all layers with consistent reporting, evidence that the layer caused avalanches, and more than 10 persistent avalanche problem days. Class 2 (Good) represents layers that were still of operational concern, but either the reporting was slightly inconsistent, or the layer was associated with a persistent problem for less than 10 days, or the comments showed that the layer was rather unreactive to triggering. Class 3 (Uncertain) contains

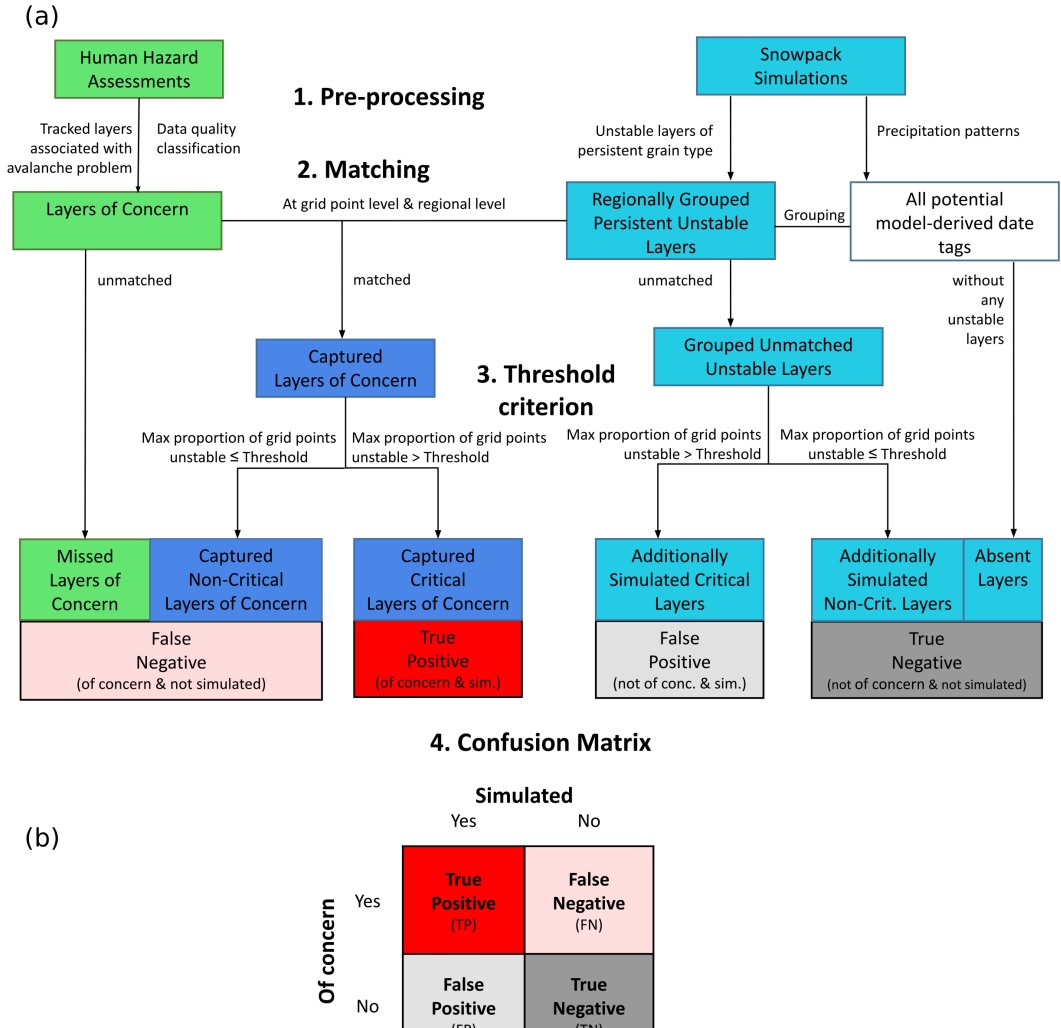

**Figure 4.** (a) A flowchart for assembling the four cells of the confusion matrix from the human and modeled data sets. The human data set is displayed in green boxes, snowpack simulations in turquoise, data points informed by both data sets in dark blue. (b) The resulting confusion matrix categorizes all critical layers into two dimensions: whether they are of human concern and whether they are identified by the simulation. All processes visualized by this figure are explained in detail in Sect. 3.1.1–3.1.5

.

layers whose assessments hold substantial uncertainty. While the assessments suggest that these layers existed, they were only linked to very few storm or persistent avalanche problems, showed very inconsistent reporting, or were of only limited

operational concern due to missing avalanche reactivity. Class 4 (Unsure) consists of layers that were barely reported upon, making it questionable whether they existed at all.

### 3.1.2 Pre-processing simulations

To identify relevant layers in the snowpack simulations, we searched all simulated snow profiles for layers that were characterized by persistent grain types and poor stability. Persistent grain types included all faceted layers (FC, DH), surface hoar layers

(SH), and crust layers (MFcr, IF). Since crust layers are strong layers, we counted crust layers as unstable if unstable weak layers (FC) were present in the vicinity of the crusts. We employed two different approaches to assess dry snow instability: a recently developed statistical approach and a process-based approach based on our current understanding of avalanche release. Since the existing literature does not yet provide any guidance on which of these approaches is superior or more appropriate for our application, we evaluated the entire confusion matrix analysis described in Sect. 3.1 twice, once employing the statis-

tical approach and once the process-based approach. To improve readability of our manuscript, layers or grid points with poor stability are also referred to as unstable layers or grid points.

The statistical approach used the random forest classifier developed by Mayer et al. (2022). This model was trained with a high-quality data set of observed snow profiles recorded around Davos, Switzerland. Based on the observed instability of the weakest layer in each profile, the model learned to predict the probability of layer instability ($p_{unstable}$) from a set of six

simulated predictor variables. These predictor variables included characteristics of both the weak layer and the slab, namely the viscous deformation rate, the critical cut length, the sphericity and grain size of the weak layer, the skier penetration depth, and the cohesion of the slab. As suggested by Mayer et al. (2022), we considered layers with $p_{unstable} \geq 77$ % as critical avalanche layers with poor stability.

Inspired by the work of Monti et al. (2014) and Reuter et al. (2021), our process-based approach used a combination of three

indices to assess the instability of a layer: a) the relative threshold sum approach RTA (Monti and Schweizer, 2013; Monti et al., 2014), b) the multi-layered skier stability index SK38 (Monti et al., 2016), and c) the critical crack length $r_c$ (Richter et al., 2019). Each of these indices consists of a variety of weak layer and slab characteristics, such as macroscopic properties (e.g., layer depth), microstructural properties (e.g., grain type and size), and mechanical properties (e.g., shear strength). While potential weak layers were pre-selected based on RTA, their propensity for failure initiation and crack propagation was assessed

based on SK38 and $r_c$, respectively, which accounts for the two main processes governing slab avalanche release (Schweizer et al., 2003, 2016). While the literature agrees on thresholds for RTA and SK38, less consensus exists for $r_c$. Hence, we applied two thresholds to $r_c$ and therefore identified critical avalanche layers with poor stability if RTA $\geq 0.8$ (Monti and Schweizer, 2013), SK38 $\leq 1$ (Reuter et al., 2021), and $r_c \leq 0.3$ (Reuter et al., 2021) or $r_c \leq 0.4$ (Reuter et al., 2015).

While the relevant layers of human concern represent regional assessments with a single date tag (Sect. 2.3), the layers

identified in the simulations consist of observations from different grid points and at different times that have not been connected yet. To make the simulated layers comparable to the regional layers of concern, we grouped the simulated layers using

model-derived date tags. These model-derived date tags represent regional markers for times when layers with the potential to become critical avalanche layers got buried. Analogously to human date tags, these regional markers are primarily represented by the onset dates of pronounced storms with substantial snowfall.

To establish the model-derived date tags, we first highlighted dry periods by identifying all days with less than a trace amount of snowfall or rain for each season and region. The actual model-derived date tags are defined by the start of substantial storm periods following dry periods. However, since potentially critical layers can also be buried by small amounts of snow over multiple days, we inserted additional date tags in between these main storm periods if the non-storm periods were sufficiently long and characterized by substantial accumulations of small daily snowfall amounts. The exact rules and thresholds for

establishing the model-derived date tags are described in detail in the Appendix A.

Once the model-derived date tags were established, we assigned each simulated unstable layer to the closest date tag older than the layer's formation date. This means that all the layers associated with the same date tag can be found within a narrow band of the snow stratigraphy that got buried close to the associated date tag. This set of rules allowed us to meaningfully group simulated layers across each region and season based on precipitation patterns and their formation and burial times similarly

to how forecasters label layers of concern.

In the final step, we determined the grain type class of each group of simulated layers. To label the layer groups with either a single grain type (FC, SH, or MFcr) or by a mix of two classes (e.g., SH/FC), we first identified the two most prevalent grain types within the group of layers associated with a date tag at each grid point for each day. This allowed us to identify the two most prevalent grain types across all grid points for each day, and eventually for the entire lifetime of the layer. To address the

well-known SNOWPACK behaviour of transforming most SH layers into DH layers after they have been buried for several days to weeks, we judged DH layers as FC layers if only a negligible amount (less than 10%) of SH was encountered during the layer's lifetime. If the fraction of SH layers was higher than 10 %, we judged all DH layers as SH layers. Finally the layer groups were labeled with the resulting one or two prevalent grain types.

### 3.1.3 Matching of layers of concern and simulated layers: How and when?

The matching of layers between the human data set and the simulations is arguably the most important step in the derivation of the confusion matrix, because it influences all four cells of the matrix. To ensure that all layers were classified meaningfully, our matching approach consisted of two steps. We first identified layers of human concern at individual grid points before checking at the regional level to ensure we did not miss any relevant layers based on the grouping of layers by model-derived date tags. The following paragraphs explain these two steps in detail.

To identify layers of human concern in the simulated profiles at individual grid points, we constructed search windows around the human date tags of individual layers of concern. SNOWPACK creates labels that store the date and time when a particular layer is formed or deposited. Following Richter et al. (2019), we also computed burial date labels based on the deposition date label of the overlying layer. To account for biases in the reported date tags and errors in the simulated precipitation patterns, a simulated layer was considered matched if it formed within the formation window or got buried within the burial window of a

layer of human concern. The formation window ranged from the last day of the prior storm event to the day of the human date

tag, and the burial window ranged from one day prior of the human date tag to the first day of the actually simulated storm event (Fig. 5). We used thresholds for cumulative amounts of new snow (10 cm) and liquid precipitation (5 mm) to identify the end of the preceding storm event for the start of the formation window (accumulating backward) and to identify the start of the next snowfall event for the end of the burial window (accumulating forward). To avoid unreasonable matches, we limited the length of the formation window to 30 days and the length of the burial window to five days.

While the search windows identified all simulated layers that align with a specific human date tag, the layers still needed to be filtered for the reported grain types to produce a meaningful match. To accomplish this, we employed up to four different grain type searches for each layer of concern: a) a strict grain type search that only accepted the specific *reported* weak grain types (e.g., SH, SH/FC, etc.) in the simulated layers; b) a relaxed grain type search that accepted *all* persistent weak grain types (SH/DH/FC) in the simulated layers; and a grain type search for c) any crust layer (IF, MFcr) or d) any crust layer with an adjacent unstable weak layer if crusts were reported to play a role in the given layer of concern.

The final consideration for the matching of the layers was the timing of the matching. Since persistent layers persist for a long time, there are several possible options for when to match the layers from the two data sets (e.g., at their time of burial, first concern, most concern). To account for temporal variability in the presence and instability of these persistent layers, the matching routine was run daily from 5 days before the reported date tag up to 90 days after. All layers of concern that were matched to simulated layers at any time within this 96-day validation window were considered *captured*.

As described above, the first step of the layer matching routine requires a threshold for the accumulated precipitation amounts to identify the formation and burial windows. We purposely chose a low threshold to ensure that only layers within a narrow band of the snow stratigrahy were matched. However, our initial explorations revealed that approximately 5–10 % of unstable layers were erroneously assigned to model-derived date tags that were only a few days apart from the relevant human date tag. This situation can occur at grid points that either experienced a slightly lagged arrival of the storm producing later model-derived date tags or above-average precipitation amounts which results in earlier model-derived date tags. Since these layers are actually part of the same group of layers that were assigned to the correct human date tag, we changed the assignment from the model-derived date tag to the human date tag for these layers if a) the date tags were only one day apart regardless of snowfall amounts or b) the date tags were up to three days apart and no more than 10 cm of new snow was simulated in between.

### 3.1.4 Applying a threshold criterion to fill the confusion matrix

After the pre-processing and matching of the data sets, we can compute the counts for each cell of the confusion matrix (Fig. 4a). All layers of concern that remained unmatched by the simulations were counted as missed layers of concern and directly contributed to the False Negative cell of the matrix. Analogously, all potential model-derived date tags without any simulated unstable layers (rightmost arrow in Fig. 4, Sect. 3.1.2) were counted as absent layers and directly contributed to the True Negative cell of the matrix. While these missed layers of concern and absent layers conceptually only exist at the regional scale and do not have a direct link to the simulations at individual grid points, the captured layers of concern and the remaining grouped unmatched unstable layers are linked to the simulations at grid points, and we can calculate the daily proportion

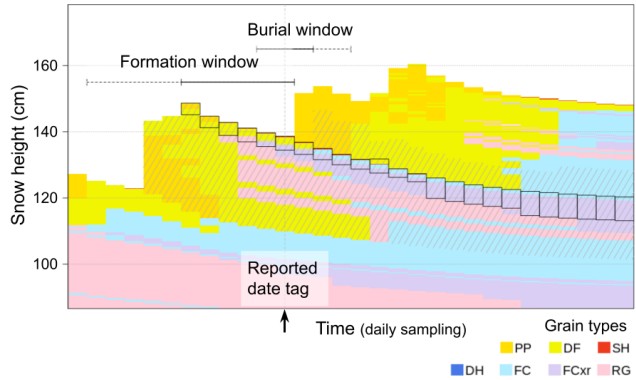

**Figure 5.** An illustration of the search windows for layers of human concern around the human date tag. The panel zooms in on the near-surface layers of a time series of a simulated snow profile (daily time step). The extent of the formation and burial windows are highlighted by horizontal lines, where the solid lines indicate the extent of each window based on the timing of the storms. The black boxes highlight all layers that either formed within the formation window or got buried within the burial window. To better illustrate the effect that different lengths of (potentially fixed) time windows have on the selection of layers, the gray hatched areas highlight layers that would result from time windows extended by the dashed horizontal lines. Colors refer to snow grain types; acronyms for these grain types are defined in Sect. 2.2 and Table A1.

of unstable grid points, which we call the *proportion unstable*. To translate this continuous variable into a binary variable and decide whether a layer should be counted as simulated (or not), we applied a threshold criterion. If the maximum daily proportion unstable within the 96-day validation window was greater than the threshold, the layer was counted as simulated, otherwise as not simulated. Hence, the captured layers of concern either added to the False Negative cell or informed the True Positive cell. Likewise, the grouped unmatched unstable layers either added to the True Negative cell or informed the False

Positive cell.

To summarize our methodology, we illustrate the concepts applied so far with the 2019 winter season in GNP at TL (Fig. 6). The time series of the average profile (Herla et al., 2022) for the region, shown in the bottom left panel, provides context for the date tags that mark the beginning of important snowfall periods (Sect. 2.3, 3.1.2, 3.1.3). Date tags that were reported by forecasters are visualized by solid red vertical lines while the additional model-derived date tags are indicated by dashed gray

vertical lines. For each regional layer represented by its date tag, a bar in the bar chart of the top left panel represents the maximum daily proportion unstable. Using 50 % as our threshold criterion for this example, each bar is colored according to its corresponding cell in the confusion matrix in the bottom right panel. The violin plots shown in the top right panel present the distributions of the maximum daily proportion unstable in more detail, while also adhering to the same colors. In this particular case, all six layers of concern (represented by the red bars, red vertical lines, red violin, and dark red cell of the

confusion matrix) were well captured, and all the other simulated layers (represented by all gray features) generally had lower proportions of unstable grid points. However, there are five layers that were considered critical by the model using the 50 % threshold but not the human forecasters.

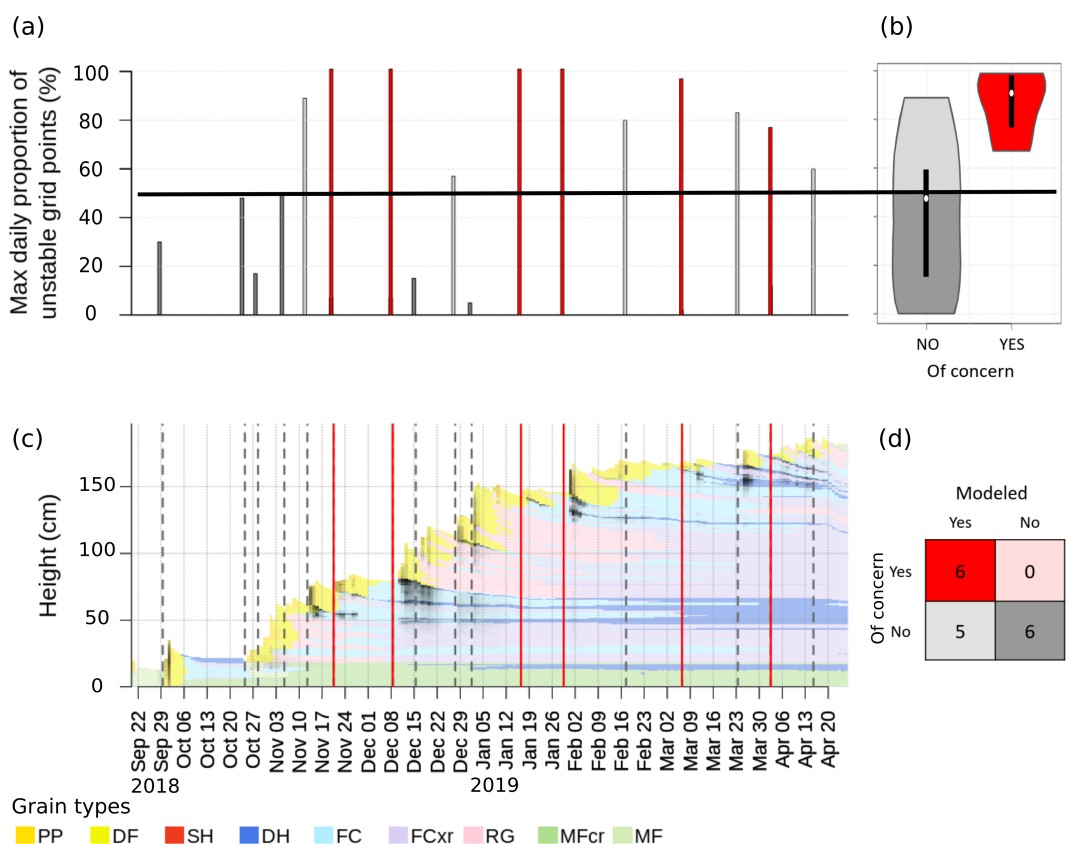

**Figure 6.** An illustration of the concepts applied to a case example of the 2019 season in Glacier National Park (GNP) at treeline elevation (TL). (a) Maximum daily proportion unstable for each regional layer (red: of human concern, gray: not of human concern) and an exemplary threshold criterion of 50 % (black horizontal line; shades of gray bars depend on the threshold criterion: dark gray corresponds to 'not modeled', light gray corresponds to 'modeled') and (b) the resulting distributions (colors are shared with panels a and d; boxplots within violins represent the median and interquartile range). (c) The average profile for the region and season (only shown to improve context; black shading highlights times and layers of modeled instability) with the human and model-derived date tags (red and gray vertical lines, respectively). (d) The confusion matrix evaluated for this specific example. Acronyms of snow grain types listed in the legend are defined in Sect. 2.2 and Table A1.

### 3.1.5 Statistical performance measures

According to (Wilks, 2019, p. 374f), three complementary attributes are required to comprehensively capture the model performance. We present the probability of layer detection, the model precision, and the false alarm rate. Probability of detection, also known as recall or model sensitivity, is the ratio of True Positives / (True Positives + False Negatives), which describes the probability that a layer of concern is captured by the simulations. This addresses the first practical forecaster question we outlined in our research objectives. Model precision, also called positive predictive value, is the ratio of True Positives / (True Positives + False Positives). It represents the probability that a modeled critical layer is indeed of operational concern, which corresponds to the second practical forecaster question. Finally, the false alarm rate, also referred to as 1 - model specificity, is the ratio of False Positives / (False Positives + True Negatives). It expresses the probability that a layer that the human forecasters are not concerned about is identified as a critical layer by the model.

To present our results, we computed the three performance attributes for all possible threshold criteria between 0–100 %. The attributes were then visualized in a precision–recall curve and a relative operating characteristics (ROC, historically also known as receiver operating characteristics) curve using the software package 'ROCR' by Sing et al. (2005) written for the R Language and Environment for Statistical Computing (R Core Team, 2020). Each point on these curves describes the model performance for one specific value of the threshold criterion, which can then be compared against a no-skill baseline. The no-skill baseline represents the performance of a random model in the ROC curve, and the climatological frequency of critical layer formation in the precision–recall curve. To account for uncertainty in the assessment of model performance, we computed 95 % confidence bands around each curve by bootstrapping the confusion matrix results at each threshold following Davison and Hinkley (1997) and using the R package 'boot' (Canty and Ripley, 2022). To provide insight on the effect of grain type, elevation band and forecast region on performance, we stratified the overall model performance by these variables and visualized them in additional precision–recall and ROC curves.

In addition to the three performance attributes, we computed two skill scores, the Peirce skill score (PSS) and the $F_1$ score, that summarize the different aspects of model performance in a single measure. Both scores express model performance relative to a perfect model using different ways of combining the cells of the confusion matrix. While these scores are less tangible to interpret, they simplify comparing model performance at different threshold criteria and for different approaches of modeling instability. The PSS, also known as the true skill statistic, accounts for all cells of the confusion matrix to express the actual skill of the simulations (Eq. (1)), whereas the $F_1$ score disregards the True Negative cell and is therefore biased towards the prediction of events (Eq. (2)). Since the prediction rate of rare events can be high simply by chance, PSS is the preferred score for our application (Ebert and Milne, 2022). Despite this limitation of $F_1$, it provides a complementary view on model performance that is independent of our approach to compute the True Negative cell based on model-derived date tags. For more information on skill scores in rare event forecasting please refer to Ebert and Milne (2022) who provide a comprehensive overview and discussion.

The PSS and $F_1$ scores are defined as

$$PSS = \frac{(TP \cdot TN) - (FP \cdot FN)}{(TP + FN) \cdot (FP + TN)} \tag{1}$$

$$F_1 = \frac{2TP}{2TP + FP + FN} \tag{2}$$

where TP and TF are true positive and true negative counts, and FP and FN are false positive and false negative counts, respectively (Fig. 4).

## 3.2   Finding more detailed patterns in all missed or captured layers of concern

The next components of our persistent layer validation analyze all missed or captured layers of human concern in more detail. This section describes how we searched for patterns in these two groups of layers to better understand the subtleties of model performance.

While the confusion matrix analysis described in the last section only included layers with poor stability, the analysis
described here also includes layers of concern that where captured structurally but remained stable. Analogous to our approach of using *proportion unstable* to characterize relevant layers in the last section, we use *proportion captured* to characterize the mere structural presence of each missed or captured layer independently of its simulated stability. The computation and analysis of the proportion captured follows the same steps as the analysis of the proportion unstable as illustrated in Fig. 4 except that it skips the first pre-processing step of filtering for unstable layers. Identical to the confusion matrix analysis, we computed
the maximum daily values for the proportion captured and proportion unstable for the 96-day validation window. By including both the proportion captured and the proportion unstable, we acknowledge that the simulations first need to structurally capture the layers of concern before they can provide insight about instability.

To provide insight on what layer attributes and contextual factors influenced the maximum proportion captured and unstable, we used conditional inference trees (CTree) (Hothorn et al., 2006), a type of classification tree that uses a statistical criterion for
finding splits. CTrees recursively partition the distribution of a response variable based on the statistically most significant splits along a set of explanatory variables. For the present analysis, we used the ctree function in the R package 'partykit' (Hothorn and Zeileis, 2015). Due to their ease of use and easily interpretable results, CTrees have already been used several times in snow and avalanche research for exploratory analyses (e.g. Horton et al., 2020b). While the top node of a CTree represents the most significant split that divides the entire sample, the resulting subsamples are recursively split into smaller subsamples until the
algorithm cannot find any significant splits in the the response variable anymore. The resulting terminal nodes describe subsets of the data set with distinct distributions of the response variable that can be linked to specific combinations and thresholds of the explanatory variables.

We included the following potential explanatory variables in our analyses, which contained both direct layer attributes as well as more contextual information:

– grain type (and grain type search routine)

– grain size of the simulated layer at burial

- data quality class

- month of burial

- number of associated avalanche problem days

- simulated length of the dry spell before burial

- season

- region and elevation band.

Since the CTree analysis required the explanatory variables to remain constant over the lifetime of the layer, we did not include any other specific weak layer or slab variables.

### 3.3 Estimating agreement indicators between modeled and reported instability of captured layers of concern


The analysis described so far validated the existence and instability of persistent weak layers and crusts purely based on the assumption that layers tracked by forecasters were of operational concern at some point during the seasons. However, the daily assessments of the avalanche problems according to the conceptual model of avalanche hazard provide more detailed information about the associated layers. In particular, the reported likelihood of a persistent avalanche problem is closely
related to the instability of the linked layer of concern.

To assess the quality of the simulated unstable layers in more detail, we compared the layers' simulated proportion unstable to the reported likelihood of the associated persistent avalanche problem. Forecasters expressed the likelihood of an avalanche problem on a five level ordinal scale ranging from *unlikely*, over *possible*, *likely*, *very likely*, to *almost certain* (Statham et al., 2018a). To quantify the agreement in layer instability $\Psi$, we defined several agreement indicators related to the variation and
timing of instability. For variation, we used the Spearman rank correlation ($\rho_\Psi$) (Wilks, 2019, p. 59f) between the proportion unstable and the assessed likelihood. $\rho_\Psi$ ranges between [-1, 1] with a perfect positive correlation at 1 and no correlation at 0. For the timing we used (i) the lag in the onset of the layer starting to be a problem ($\Lambda_{onset}$), (ii) the lag between the layer ceasing to be an issue in the simulations and forecasters removing the associate avalanche problem in the bulletin ($\Lambda_{turn-off}$), and (iii) the difference in the total number of days that the layer appeared to be a problem ($\Delta_{duration}$). To quantify the relevant times in
the simulations we used the proportion unstable with different thresholds. After initial explorations with 5, 20, and 50 % for the threshold, we settled on using 20 % for all calculations.

To offer insights into possible reasons for why certain layers were modeled better than others, we performed a series of CTree analyses on the individual indicators similar to how we used CTrees in the previous section. To focus this analysis, we reduced our data set to layers of concern at TL and layers for which all agreement indicators could be calculated. For example,
we removed all layers that were characterized by a constant likelihood, which prevented calculating the correlation coefficient $\rho_\Psi$. We also removed all layers whose maximum proportion unstable was lower than the chosen threshold for $\Lambda$. For a threshold of 20 %, for example, the resulting data set contained 96 layers of concern.

## 4 Results

### 4.1 Inventory of all layers of concern

Our database contained a total of 167 layers of concern, of which 107 layers were identified to be primarily of weak grain type (SH, DH, FC) and 60 to be primarily crust layers (MFcr, IF). Most layers of concern were reported and tracked in GNP (on average 8.4 layers per season), followed by BYK and then S2S (on average 4.7 and 2.2 layers per season, respectively) (Fig. 7b, d, f). While GNP and BYK reported twice as many weak layers than crust layers, S2S reported an equal ratio. The data quality of the layers was assessed as excellent or good in 50–60 % of cases. The detailed annual inventory of all layers of concern revealed a high inter-seasonal variability in the number of reported layers without a long-term trend (Fig. 7a, c, e).

The number of days with persistent avalanche problems showed a slightly different pattern (solid lines in Fig. 7a, c, e). While S2S experienced persistent avalanche problems at about 30 % of all days of forecasting operations, 60 and 70 % of the days in GNP and BYK had a persistent avalanche problem. In S2S and GNP the vast majority of issued persistent problems were linked to tracked layers of concern, whereas in BYK these problems were linked to tracked layers in only 70 % of the cases (dotted lines in Fig. 7b, d, f). The number of persistent problem days fluctuated substantially from year to year in S2S and BYK (Fig. 7a, e). Although the number of persistent problem days was more steady in GNP, two seasons deviate substantially from the seasonal average (Fig. 7c).

### 4.2 General simulation performance

We first discuss the overall performance of the simulations for representing layers of weak grain types (SH, DH, FC) using the statistical approach of simulating layer instability for all regions, seasons, and elevation bands combined. The maximum daily proportion of unstable grid points obtains significantly different distributions for layers that were of operational concern and for those that were not (Wilcoxon rank sum test: P < 0.001, Fig. 8). Most layers of concern tend to be associated with more unstable grid points than layers that were not of concern. The resulting performance curve shows a linear decrease of precision with increasing probability of detection (Fig. 9a). As expected, the maximum precision of 60 % is attained with a high threshold for proportion unstable of 95 %, and precision drops slightly under 40 % for proportions unstable below 20 %. In contrast, the probability of detection increases with decreasing proportion unstable and maxes out slightly above 80 % (Fig. 9a, b). The false alarm rate increases with decreasing proportion unstable, first slowly for high proportions and then stronger for low proportions unstable. The precision–recall and ROC curves both indicate that the simulations have skill in capturing layers of concern, but their performance is far from perfect. A perfect model would produce a horizontal line at the top of the precision–recall curve, and the line in the ROC curve would follow the y-axis to the upper left corner and then across at the top of the chart. On the other side, the no-skill base line is represented by a horizontal line in the lower part of the precision–recall curve and by the diagonal in the ROC curve (dashed lines in Fig. 9).

Analyzing the overall model performance with the process-based approach to assessing layer instability reveals several findings. Overall, the process-based approach yields better results with a threshold for the critical crack length of $r_c \leq 0.3$ than $r_c \leq 0.4$ (Fig. 9c, d). The performance of the process-based approach with the higher threshold is significantly lower

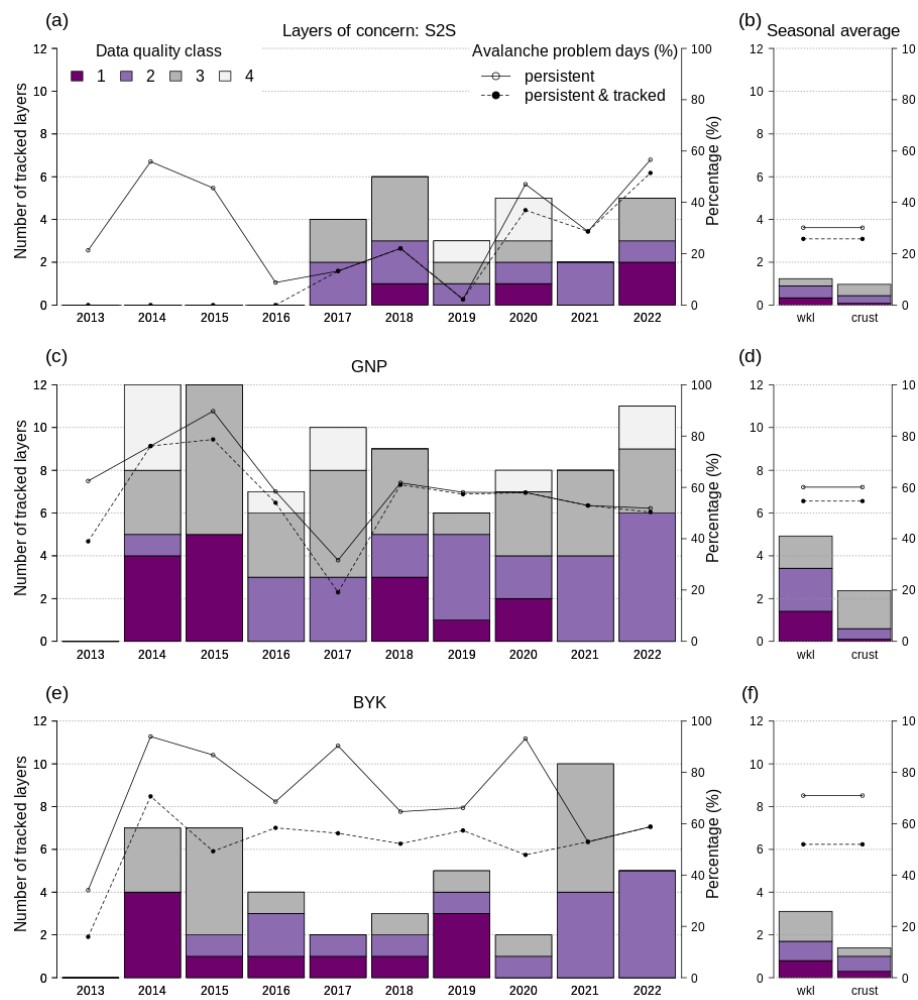

**Figure 7.** An inventory of tracked layers of concern that were extracted from the human hazard assessments for the regions Sea-to-Sky (S2S), Glacier National Park (GNP) and Banff-Yoho-Kooteney (BYK). Detailed inter-seasonal patterns are visualized for each region in (a, c, e), and seasonal averages are shown in (b, d, f) for weak layers (wkl) and crusts. The colored bars sum up the number of layers in each data quality class, and the lines show the percentage of days per season where a persistent avalanche problem was issued (solid line and open circles), and linked to a tracked layer of concern (dashed line and filled circles).

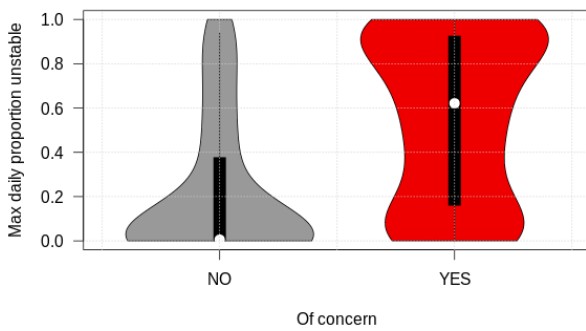

**Figure 8.** The overall distributions of the maximum daily proportion unstable for all weak layers (surface hoar/depth hoar/facets) that were and were not of operational concern.

than the statistical approach on the 95 % confidence level. The process-based approach with the lower threshold, however, yields comparable results to the statistical approach and lies within the confidence band of the statistical approach (Fig. 7a, c).. Furthermore, while the process-based precision–recall curve with the higher threshold remains constant at approximately 40 % for all proportions unstable, the process-based approach with the lower threshold follows a similar slope as the statistical approach and reaches a comparable maximum precision. At equal proportions unstable both process-based curves show a 10-15 percentage points lower probability of detection than the statistical approach, which can be seen in all panels of Fig. 9. Lastly, the false alarm rate of the process-based approach with the lower threshold follows the same pattern of the statistical approach, but at identical proportions unstable the process-based approach yields an approximately 8 percentage points lower false alarm rate.

Condensing the three performance attributes into the Peirce skill score (PSS) reveals a very flat peak of the statistical approach at a threshold of approximately 20 % for the proportion of unstable grid points and a PSS of 0.41 (Fig. 10a). The process-based approach with the lower threshold obtains a maximum PSS that is five percentage points higher, albeit only for very low thresholds of the maximum proportion unstable of around 3 %. For higher proportions the PSS of the process-based approach decreases linearly below the PSS of the statistical approach. The $F_1$ curves show a similar pattern, but with an even flatter peak and a slower decrease with increasing proportions unstable. For our subsequent analysis, we focus on the statistical approach and neglect the process-based approach.

Calculating separate performance curves for grain types revealed that SH layers are modeled with a precision between 80-100 %, whereas FC layers only between 25-40 %. False alarm rates are also significantly lower for SH layers. The probability of detection, however, is slightly better for FC (Fig. 11a, b). Crust layers are poorly represented in the simulations with probabilities of detection below 50 % at a low precision of 10-20%. All performance measures for crusts closely follow the no-skill base line (Fig. 11a, b). To make the stratification by elevation bands and forecast regions more insightful, we computed panels (c–f) without the influence of crust layers. The performance curves for the different elevation bands highlight that the simu-

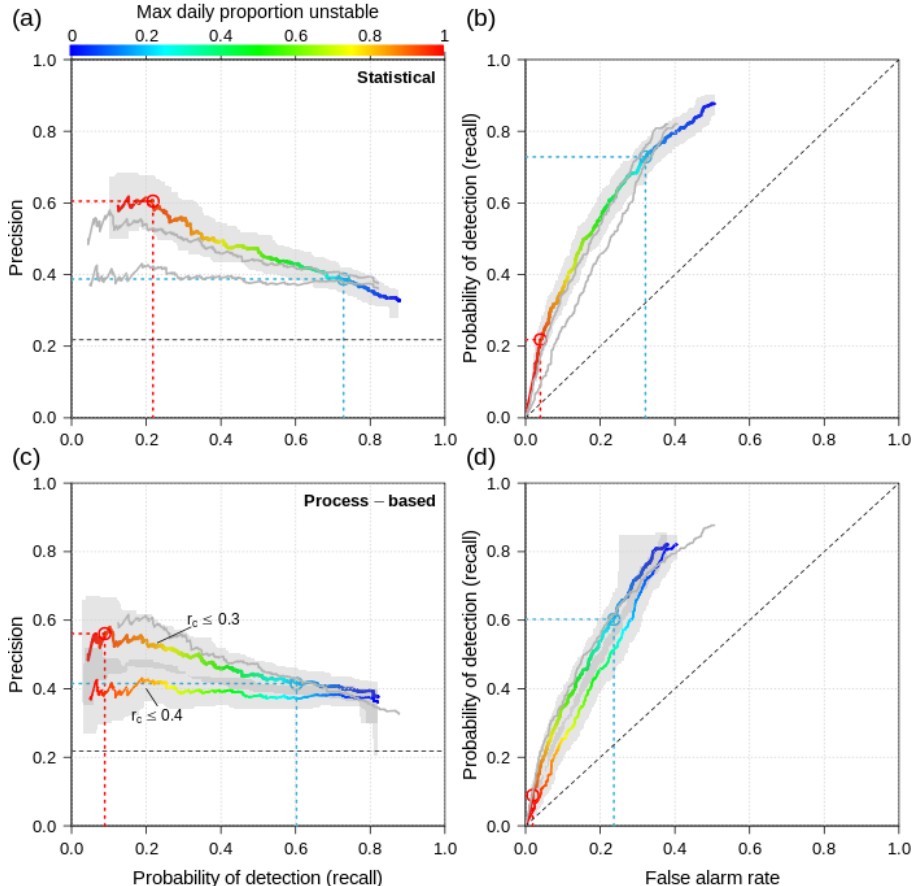

**Figure 9.** (a), (c) Precision–recall and (b), (d) ROC curves characterizing overall model performance for weak grain types in all forecast regions and at all elevation bands given different thresholds of the maximum daily proportion unstable (color). (a), (b) for the statistical approach of simulating layer instability, and (c), (d) for the process-based approach employing two different thresholds for the critical crack length $r_c$. Gray bands indicate 95 % confidence bands, gray lines shadow the curves of the other panel.

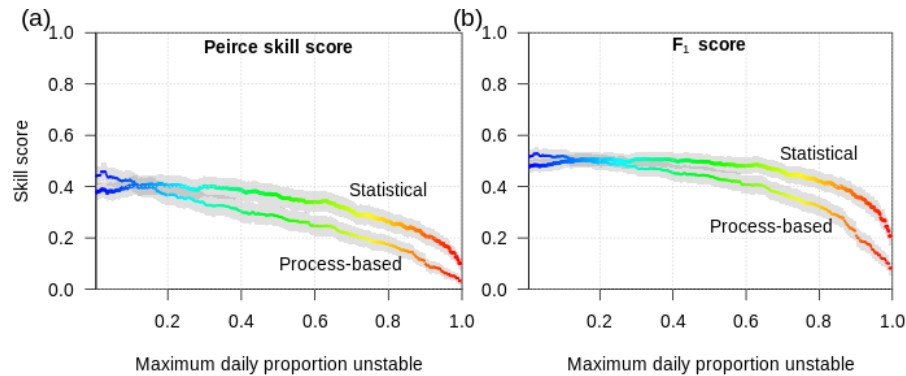

**Figure 10.** Skill scores for the statistical and process-based approaches to modeling layer instability as a function of the threshold for maximum daily proportion of unstable grid points. The process-based approach in this curve uses a threshold of the critical crack length $r_c \leq 0.3$. Gray confidence bands represent the 95 % level. Color refers to the same value as the x-axis and is shown only to facilitate cross-referencing to Fig. 9.

lations at TL score substantially better than in the ALP with higher precision at higher proportions unstable and a generally slightly better probability of detection (Fig. 11c, d). False alarm rates remain comparable. The performance curves for BTL is
between the TL and ALP, but more closely aligned with TL. The curves for the different forecast region show that precision is substantially higher in GNP (50-90 %) than in S2S and BYK (20-50 %), while the probability of detection is identical or slightly lower in GNP (Fig. 11e, f). False alarm rates are also lower in GNP than in S2S and BYK, although the effect is not as pronounced as for precision. The confidence bands highlight a substantially higher uncertainty in S2S than in BYK and than in GNP, which is due to differences in sample sizes.

### 4.3  Patterns in representing layers of concern

#### 4.3.1  Missed or structurally captured layers of concern

The CTree analysis examining the differences between layers that were structurally captured and those that were not revealed that the proportion captured most significantly depends on the grain type (Node 1–first split of the CTree, Fig. 12). While sun crusts (IFsc) are virtually absent from our simulations (Node 13), rain crusts and temperature crusts (IFrc, MFcr) are more
likely to be captured, particularly when they formed in the early season (Nodes 11 and 12). The proportion captured of SH layers is substantially influenced by the length of the dry spell before the layer got buried. While the majority of SH layers that got buried after a dry spell of at least seven days are structurally captured by our simulations (Node 7), the proportion captured varies widely within the entire spectrum if the dry spell is shorter than seven days (Node 8). In contrast to crusts and SH layers, all FC layers of concern are structurally present in the large majority of grid points (Nodes 4 and 5), although some FC layers
of lower data quality class are represented by lower proportions captured (Node 5).

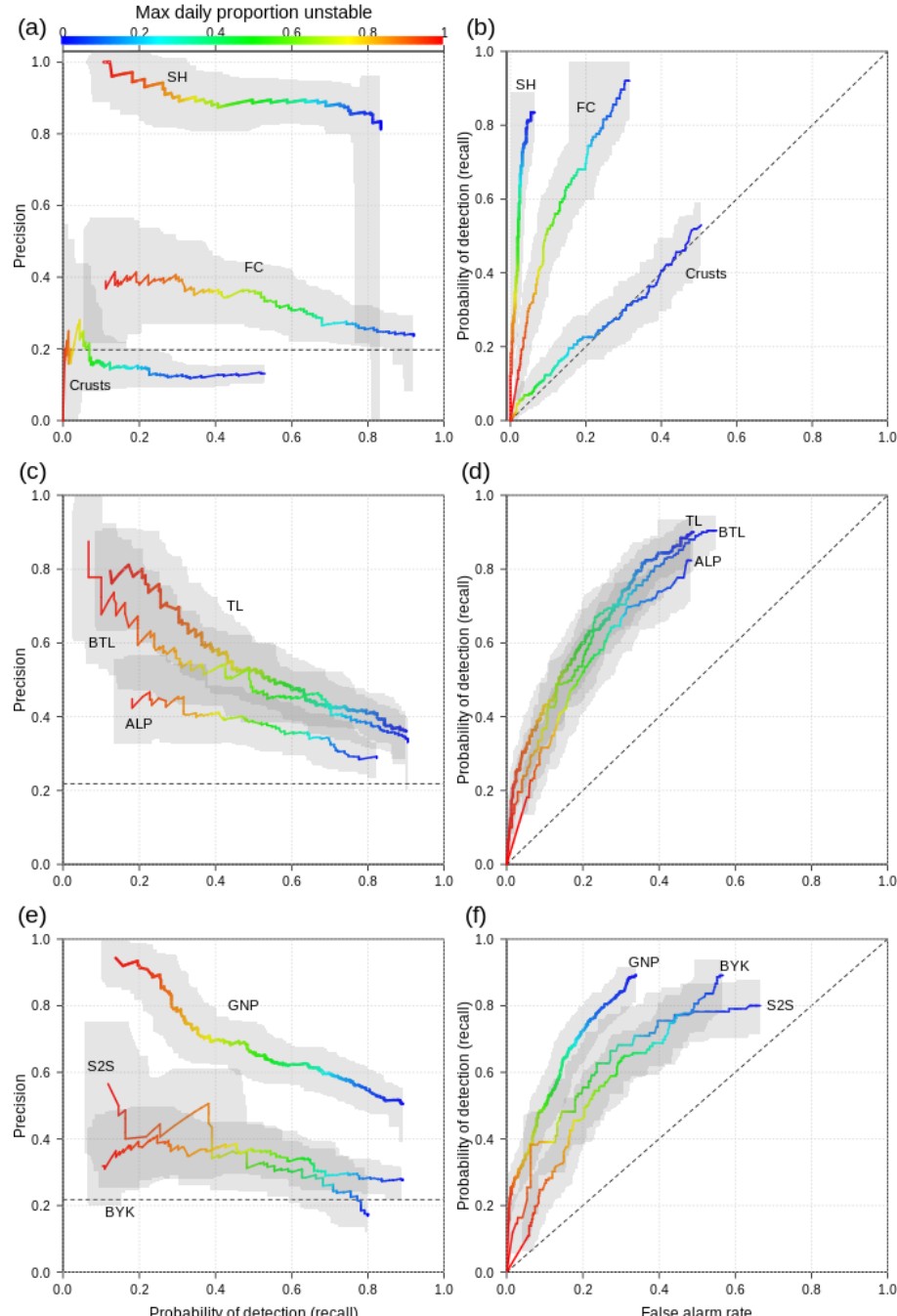

**Figure 11.** (a), (c), (e) Precision–recall and (b), (d), (f) ROC curves characterizing model performance for different grain types, elevation bands, and forecast regions given different thresholds of the maximum daily proportion unstable (color). Due to the performance of crust layers, panels (c–f) show the combined effect of SH and FC layers without the consideration of crust layers. All panels were computed with the statistical approach of modeling layer instability. Gray bands indicate 95 % confidence bands.

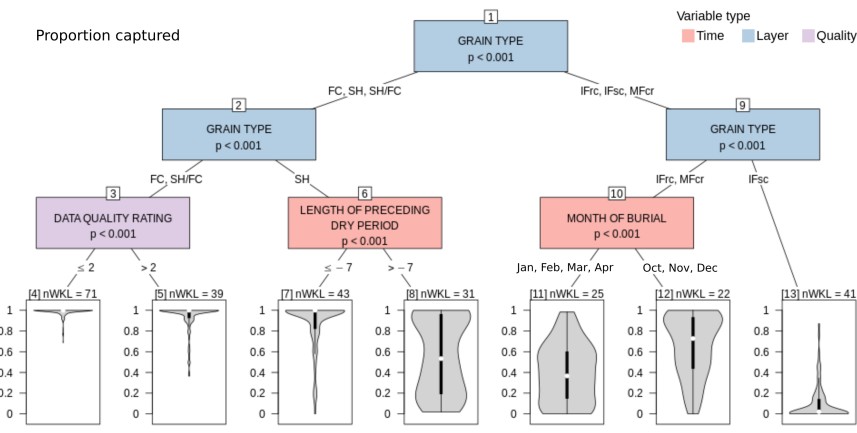

**Figure 12.** CTree for proportion captured examining the differences between layers of concern that were structurally captured by many grid points and those that were captured by few grid points or none. Terminal nodes are labeled with the number of distinct layers they contain (nWKL).

While the CTree algorithm found additional significant splits in the terminal nodes shown in Figure 12, many of them became difficult to interpret and did not provide additional insight. Hence, we are not showing the full tree. However, some splits revealed patterns worth mentioning. For example, FC were almost always present next to SH in GNP and BYK even if they were not reported. Early season IFrc and MFcr (Oct, Nov, Dec) were well captured at BTL and moderately well captured at ALP and TL (albeit with large variability). Mid- and late season IFrc and MFcr were less well captured in all elevation bands.

### 4.3.2 Missed or structurally captured and unstable layers of concern

Due to the poor model performance in capturing crust layers, we limited the CTree analysis of all missed or structurally captured and unstable layers of concern to weak grain types (SH, DH, FC). Upon fitting a CTree with the full list of potential explanatory variables, we found that 'season' continuously emerged as an important significant variable, highlighting that model performance varies substantially between seasons. Seasons with good performance include 2014, 2016, and 2019, whereas 2017 and 2020 were characterized by particularly poor performances (not shown). Since the influence of 'season' made the resulting CTree challenging to interpret, we removed it from the list of potential explanatory variables for the analysis presented in Fig. 13.

After removing season effect, the governing variables were the same as in the CTree for just structurally captured layers: grain type, length of the dry spell, and data quality. Hardly any SH layers became unstable in any grid points if the length of the preceding dry spell was shorter than 7 days (Node 9). If the dry spell was longer, the proportion unstable was substantially larger for most SH layers (Node 8). Layers that consisted of FC or mixtures of SH/FC became unstable at most grid points and for most layers of concern, if the length of the dry spell exceeded eight days or was shorter but associated with a high

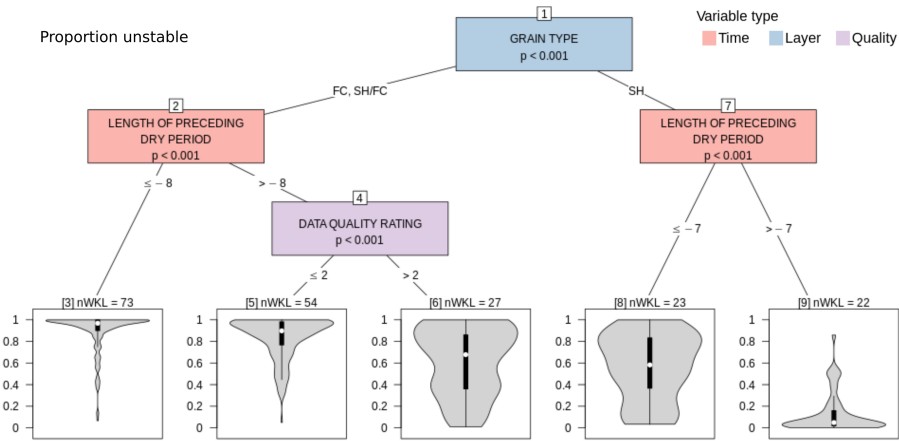

**Figure 13.** CTree for proportion unstable examining the differences between layers of concern that were modeled unstable by many grid points and those that were modeled unstable by few grid points or none. Terminal nodes are labeled with the number of distinct layers they contain (nWKL).

data quality class (Nodes 3 and 5). Layers that were characterized with a shorter dry spell and lower data quality class showed substantially more spread (Node 6).

Although the simulated grain size of SH layers at their time of burial did not show up as a significant variable in the CTrees that were fitted with multiple variables, a univariate analysis confirmed a significant relation with the proportion unstable: larger grain sizes tended to be associated with higher proportions unstable (not shown).

### 4.3.3 Agreement indicators between modeled and reported instability of captured layers of concern

The agreement in modeled and reported variation of instability $\rho_\Psi$ was primarily influenced by the number of days each layer was associated with a persistent avalanche problem (Fig. 14a). A strong negative correlation tended to be attributed to layers whose instability was very short-lived (Node 2) and can likely be attributed to phase lags of the peak instability. The best agreements were found with layers that were of concern for more than 14 days (Node 5), and among those for layers whose modeled onset of instability was close to their reported onset (not shown). The remaining layers were characterized by a highly variable agreement, many appeared to be even uncorrelated (Node 4). The agreement was not influenced by grain type or data quality class. However, a qualitative inspection of all layers that were of concern for more than 14 days but characterized by a $\rho_\Psi < 0.4$ suggested the following possible reasons for the poor agreement:

- The instability was primarily reported for crust layers, none of which were properly captured by the simulations (five layers, all in BYK)

- The reported instability tapered off while the modeled instability remained high (four layers of good data quality class)

- The modeled instability at the time of concern was dominated by a different weak layer (two layers of poor data quality class)

- The proportion unstable followed the variation of the danger rating, while the reported likelihood remained constant and therefore yielded worse correlations (two layers of good data quality class)

- A modeled precipitation event was underestimated, which led to underestimated instability in buried weak layers (two layers of good data quality class)

- The modeled instability tapered off earlier than the reported instability (one layer of good data quality class).

The agreement indicator $\Lambda_{\text{onset}}$ for the onset of the layer starting to be of concern was found to be independent of the threshold used for the analysis. Furthermore, for almost all layers the modeled onset of the instability was earlier than or at the time of the reported onset (Fig. 14b). For FC layers or SH and SH/FC layers with pre-burial dry spells longer than 14 days the onset date tended to be modeled substantially earlier than reported, often up to three weeks earlier (Nodes 2 and 4). The smallest deviation in the onset date and therefore best agreement was found for SH and SH/FC layers, whose dry spell was shorter than 14 days (Node 5).

Similarly to $\Lambda_{\text{onset}}$, the agreement indicator $\Lambda_{\text{turn-off}}$ for the timing of when layers ceased to be of concern was not affected by the threshold used. $\Lambda_{\text{turn-off}}$, however, was evenly distributed around lag zero, which means that the modeled instability sometimes healed sooner and sometimes later than the reported instability. The number of problem days was the only driver influencing the distribution (Fig. 14c): While layers that were of concern for more than 17 days tended to be modeled stable sooner (Node 3), layers that were of concern less long tended to be modeled stable later than what the human forecasts had suggested (Node 2).

Consistent with the results for the timing of when layers started and ceased to be of concern, the agreement indicator $\Lambda_{\text{duration}}$ was influenced more heavily by a potential lag of the final phase of the layers' concern than during its onset. The combined effects led to most layers being modeled unstable for more days than they were associated with avalanche problems (Fig. 14d). Only layers that were of concern for more than 16 days and were characterized by a dry spell shorter than 12 days tended to be modeled unstable for fewer days (Node 9). However, any layer associated with a long dry spell tended to be modeled unstable for more days (Nodes 3 and 8). Layers that were of concern for less than 16 days without long dry spell tended to be modeled unstable for only few more days (Nodes 5 and 6).

## 5  Discussion

### 5.1  Insights for the application of snowpack modeling in operational avalanche forecasting in Canada

Our results have shown that the Canadian weather and snowpack model chain is able to skillfully represent weak layers that are of operational concern. However, comparison against human assessments also demonstrates substantial differences between the two data sets. Both data sets are characterized by uncertainty, and both forecasts—modeled and human—are far from

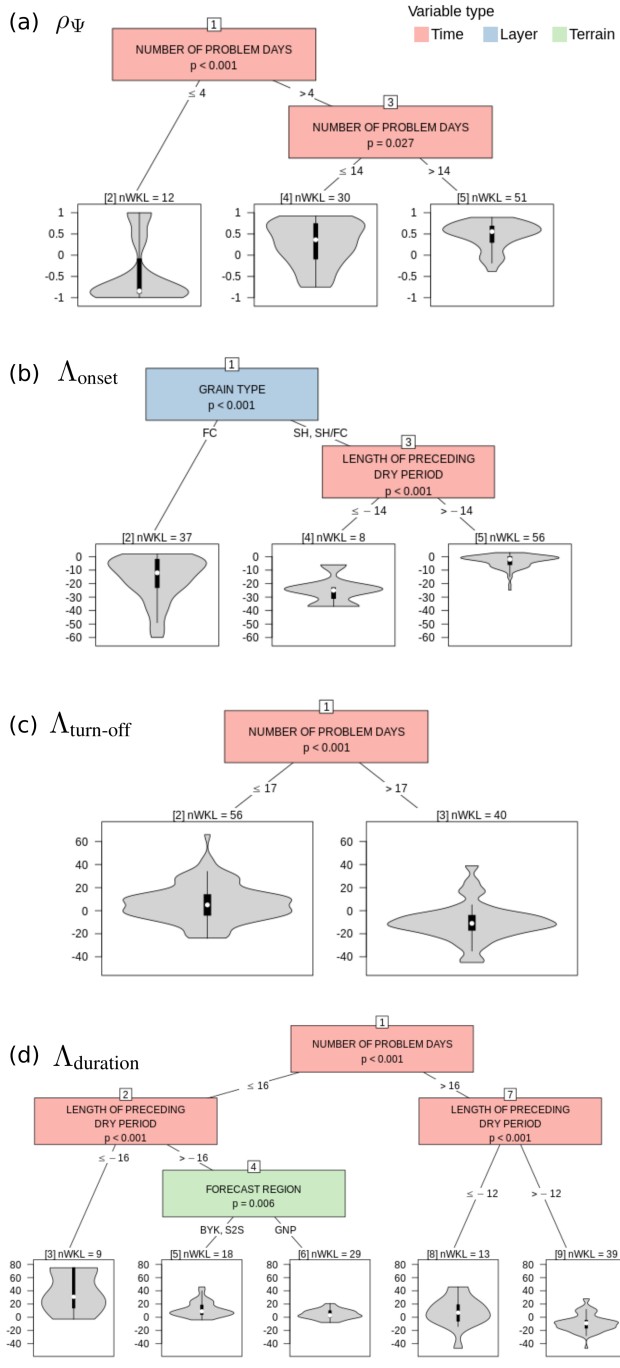

**Figure 14.** CTrees for different agreement indicators between modeled and reported instability. (a) CTree for the indicator of variation $\rho_\Psi$ examining the differences between layers of concern whose reported instability agreed well and poorly. (b), (c), (d) similar CTrees but for the indicators of timing $\Lambda_\text{onset}$, $\Lambda_\text{turn-off}$, and $\Lambda_\text{duration}$, respectively. Terminal nodes are labeled with the number of distinct layers they contain (nWKL).

perfect. While this fact is discussed in detail in the subsequent section (Sect. 5.2 Limitations), the current section interprets the results of our validation from a practical perspective of applying snowpack simulations to support avalanche forecasting in Canada. To address avalanche forecasters and snowpack modelers, who strive to develop tools for avalanche forecasters, we focus on a wide range of tangible model scenarios that cover different regions, elevation bands, grain types, and a varying degree of simulated instability, which we express as the proportion of unstable grid points within a forecast region. These take-home messages will help understand and interpret occurrences of weak layers and crusts in the current Canadian weather and snowpack model chain.

### 5.1.1 A tangible interpretation of overall model performance

When using model simulations for assessing avalanche conditions, avalanche forecasters need to decide at what proportion of unstable grid points they should be concerned about a particular simulated layer. Naturally, the simulations capture more layers of operational concern when employing a lower threshold, but they also issue more false alarms. For a higher threshold, the number of false alarms decreases substantially, but also the number of captured layers decreases and therefore more layers of concern are missed. To illustrate this trade off, let us examine an average season, which typically includes 20 date tags of which four are associated with a layer of concern. Using a threshold of 20 % for the proportion unstable, eight date tags are identified as unstable layers in the simulation, but only three are actually of human concern whereas five are not of human concern. With this threshold, only one layer of human concern (out of four) is missed (Fig. 15a). Using a higher threshold of 95 % unstable grid points, only two out of 20 date tags are associated with simulated unstable layers. Of these two simulated layers, one is identical to the layers of human concern, while one is not reported to be of concern. Three layers of human concern (out of four) are missed (Fig. 15b). These examples are informed by the more general performance results presented in Fig. 9a, b. Forecasters can use these curves to understand the interaction between the proportion of unstable grid points and the resulting model performance, which can help them make informed decisions about how to integrate the simulated information into daily decisions. The precision–recall curve is particularly important for forecasters who find unstable layers in the simulations and ask themselves whether or not to act upon them. If the encountered layer is unstable in 20 % of grid points and the forecasters decide to take the layer seriously, the probability of capturing all layers of concern is 75 % (i.e., probability of detection), while the probability that the encountered layer is actually of concern is only about 40 % (i.e., precision) (blue circle and blue dotted lines in Fig. 9a). If the forecasters decide to take only layers with as many as 95 % unstable grid points seriously, the probability of capturing all layers of concern decreases to almost 20 %, while the probability that each of those layers will actually be of concern rises to 60 % (red circle and red dotted lines in Fig. 9a). Hence, only taking layers seriously that are unstable in the majority of grid points and ignoring layers that are represented by a small proportion of grid points means missing many layers of concern. By contrast, when acknowledging any simulated layer with a date tag, even if the layer is only present in a small proportion of grid points, forecasters can be confident that they have narrowed in on a subset of layers that captures the majority of layers of concern. They can then focus on examining observational data (e.g., field observations, avalanche observations) to discern which layers are truly layers to be concerned about and which ones are false alarms. In this way the simulations provide a valuable starting point for targeted observations.

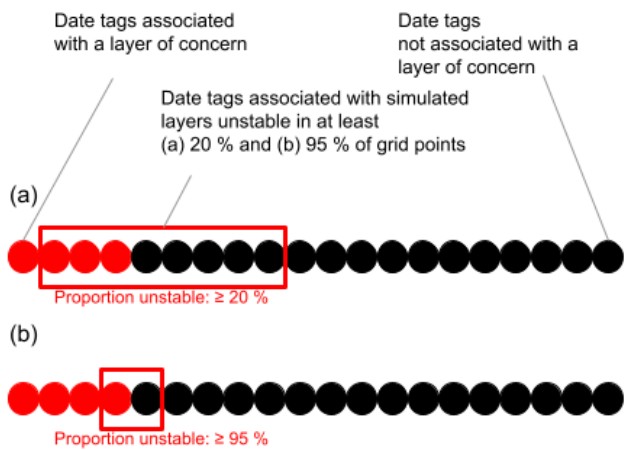

**Figure 15.** A visual representation of the confusion matrix results for different thresholds of unstable grid points. Filled circles indicate the average number of date tags per season that are associated (or not) with a layer of human concern and with a simulated layer unstable in (a) 20 % of grid points (corresponds to red circle and red dotted lines in Fig. 9a, b), and (b) 95 % of grid points (corresponds to blue circle and blue dotted lines in Fig. 9a, b).

### 5.1.2 Performance variations by grain types, elevation bands, and forecast regions

The performance of the simulation strongly depends on grain type. Simulated layers whose prevalent grain type is SH—or at least a mix of SH/FC—were very often of concern. If forecasters encounter a regionally grouped, unstable layer in the simulations that is labeled as SH or SH/FC, they can therefore assume that the layer is very likely of concern if it is unstable in more than 10 % of grid points. In contrast, pure FC layers are modeled more often than they are of concern. This causes regular false alarms from FC layers on the one hand, but on the other it causes the simulations to capture slightly more FC layers of concern than SH layers. These findings are consistent with the results of the CTree analysis, which showed that the vast majority of FC layers was structurally captured at most grid points, even if FC were not reported, and particularly if a long dry period without precipitation preceded the burial of the layer. We therefore hypothesize that our simulations tend to overpredict the structural existence and instability of FC layers.

Crust layers of concern were heavily underrepresented and underestimated in our simulations. Rain and temperature crusts of concern were sometimes structurally present by a limited number of grid points, most often in the early season and at lower elevations. Although sun crusts were regularly discussed by forecasters, they were never captured by the simulations. Since the formation of sun crusts is substantially influenced by aspect-dependent exposure to solar radiation, we believe that the lack of simulated sun crusts in our study is primarily caused by the the fact that we only used flat field simulations. Given the poor model performance in capturing any type of crust, while modeling many additional crust layers that were never reported upon, we advise to use the Canadian operational weather and snowpack model chain only for assessing weak layers (such as SH, DH, FC) and not crusts, at least until the model chain includes slope simulations and further testing has been carried out.

The performance of the simulations also varies with elevation band and forecast region. In the ALP, the model tends to produce many false alarms at any proportion of unstable grid points, while at BTL and TL layers are considerably more likely to be of concern when they are unstable in more grid points. These effects could be influenced by grain type, since there are likely less SH layers and more FC layers in the ALP than at lower elevations. The forecast region had a strong effect on model performance, which made simulations in GNP substantially more trustworthy than in S2S and BYK. This strong effect is likely

influenced by the regions' snow climates, as well as forecast practices by different agencies in different snow climates. Due to its transitional snow climate GNP is characterized by substantial snowfall amounts interspersed with frequent periods of critical layer formation (Haegeli and McClung, 2007; Shandro and Haegeli, 2018). Therefore many persistent avalanche problems exist each season, most of which can usually be linked to specific thin critical layers (Fig. 7c, d). In the maritime snow climate of S2S, critical layers form much less often and cause less persistent problems (Fig. 7a, b). Although continental BYK experiences

the most days per season with persistent avalanche problems, less persistent problems can actually be linked to specific critical layers (Fig. 7e, f). Instead of thin critical layers, the continental snowpack is often characterized by thick bulk layers of low cohesion. Since these thick layers often get deposited by different snowfall events and facet over the course of many dry spells, it can be challenging to name these layers, let alone distinguish them in the field. Our analysis approach of focusing on specific identifiable layers of concern may therefore be most applicable to GNP. For all these reasons our data set of tracked layers

of concern is skewed towards GNP and leaves BYK and particularly S2S underrepresented. Our results for S2S and BYK therefore have to be interpreted with more caution and in light of their regional peculiarities. For example, Horton and Haegeli (2022) found that BYK (and other forecast regions in the Canadian Rockies) consistently receive underestimated modeled snowfall amounts, which increases temperature gradients in the snowpack and helps explain the overestimated faceting we see in our results of BYK, which in turn leads to a low model precision in BYK. In contrast, the low model precision in S2S could

be due to an underrepresentation of layers of concern. Since instabilities are often short-lived when persistent weak layers get buried by big storms, many critical layers potentially get never associated with a persistent avalanche problem.

### 5.1.3 Detailed comparisons of human–modeled data set

Although our analysis of agreement indicators between modeled and reported instabilities likely pushed the limits of our data sets (see Limitations section, 5.2), the findings still suggest that the simulations could be used as a complementary information

source for the critical assessment of buried weak layers. Overall, most indicators that describe the timing of instability agreed on average, but we observed large variations. Furthermore, our analyses showed that forecasters tended to be concerned about layers that were associated with a persistent avalanche problem for longer than 17 days consistently longer than suggested by the simulations. This is in line with interviews with Canadian avalanche forecasters who acknowledge the challenges of taking persistent avalanche problems off the bulletin (Hordowick and Haegeli, 2022). The simulations could therefore provide

forecasters with valuable insight for when to remove persistent avalanche problems. Lastly, we found encouraging results in the indicator that describes the correlation between modeled and reported instability. The longer the layers tended to be assessed, the better the agreement turned out on average. Our more detailed qualitative follow-up analysis of layers of long-lived concern found that poor correlations were usually caused by an instability that was modeled more severe than it was assessed. In other

words, either the simulations tend to give a conservative recommendation, which is desirable, or forecasters underestimated the instability and would therefore benefit from model guidance. In other instances poor correlations could mostly be explained by other apparent reasons, such as an underestimated snowfall event, which highlights the need for continuous real-time model validation (see Sect. 5.3 for a discussion on potential future validation and monitoring).

## 5.2 Limitations

While the present results provide interesting insights for avalanche forecasters in Canada, our data sets have several limitations that need to be considered when interpreting the results. Two of these limitations have already been discussed in the previous section—the large influence of the GNP study area on the results due to the large number of persistent weak layers in a transitional snow climate, and the limitations imposed by the flat field simulations—but the primary limitation of our analysis is the fact that the validation data set consists of human assessments that do not necessarily represent an objective truth.

Our validation data set consists of human assessments that represent filtered and synthesized data from a comprehensive range of relevant observations such as weather, snowpack, and avalanche observations from different sources, locations and times. The resulting data set is available daily during the winter season and describes regional-scale conditions. There are no other data sets that describe the avalanche conditions in such a structured and consistent manner for so many seasons and over such large areas. However, since human assessments are subjective judgments that can be influenced by operational requirements and practices, they can contain biases and inconsistencies (Statham et al., 2018b; Techel et al., 2018; Horton et al., 2020b). Particularly since the Conceptual Model of Avalanche Hazard (Statham et al., 2018a) is purely qualitative and does not suggest any quantitative links between observations and hazard components (e.g., likelihood of an avalanche problem), the assessment requires substantial human judgment and expertise. While the binary assessments of layers of concern are likely one of the more straight forward and therefore reliable aspects of the used assessments, more intricate attributes such as onset or turn-off dates of avalanche problems and particularly the reported likelihood of a layer to cause avalanches are more susceptible to biases and inconsistencies.

In addition to potential human error, sparsity of observations in general can impact the quality of the human assessments. Although we chose three relatively data-rich study regions and assessed each layer of concern for its data quality to alleviate the limitations, it remains unclear whether the simulations or the human assessments were closer to reality when the two data sets disagreed. For example, while several modeled critical layers that were not reported in the assessments may be attributed to the model being overly sensitive, some could also have been missed by human observers. Similarly, while most layers of concern that were not captured by the simulations can likely be attributed to model deficiencies, a few might actually not have been as reactive as initially expected by the forecasters, or it might have been another layer that was the main cause of instability at a given day. Hence, our results can be interpreted as the lower limit of expected model performance. Lastly, any temporal comparisons between simulations and assessments are additionally impacted by uncertainties in snowfall frequency and magnitude, which in turn impact weak layer and slab characteristics alike. Again, it often remains unclear which data source is closer to reality when new snow amounts differ (Lundquist et al., 2019; Horton and Haegeli, 2022).

## 5.3 A vision for the future use of snowpack simulations in operational avalanche forecasting

Based on the insights from the present study and our practical experience with the tools developed for this research, we have developed a vision of how snowpack modeling can be embedded into operational avalanche forecasting dashboards more informatively. Despite the encouraging results found by our persistent layer validation, we do not believe that the existing model chain is sufficiently reliable to generate assessments purely based on simulations. Instead, we view the simulations as a rich complement to other information sources that can help alert forecasters about the existence of specific critical layers or provide an additional, independent perspective on their instability. To enable their full potential as a complementary information source, snowpack simulations need to be embedded in a validation suite that allows forecasters to continuously compare past assessments to past and future simulations in real time. Continuously judging the model performance will allow forecasters to develop a better understanding for the strength and weaknesses of this data source over time.

To be useful for forecasters and fit into their already busy forecasting days, such a validation suite must present the information in an intuitive way that integrates seamlessly with their existing practices (Horton et al., 2020a). Hence, we suggest to present simulated layers in a grouped format similar to layers of human concern. The methods described in this paper can be used to group and compare layers in such a format. Figures 16 and 17 illustrate how we envision the dashboards of a potential validation suite to look like using the 2019 winter season at TL in GNP as an example.

At any given day, a seasonal overview as shown in Figure 16 could show the entire history of the season and potentially the future forecast simulations for the next three days. The information presented in this view combines human assessment data with simulated information. Relevant human assessment data includes the danger rating, the tracked layers of concern, and the likelihood of their associated avalanche problems (Fig. 16a–c). Relevant simulated information includes the proportion captured, the proportion unstable, and the time series of a representative snow profile (Herla et al., 2022) (Fig. 16d, e). While the profile gives a familiar and comprehensive overview of the snow stratigraphy including snow height, new snow amounts, prevalent layers and their times of instability, the proportion captured and proportion unstable offer insight on how many grid points contain the current layer of human concern or suggest instability in that layer (colored circles and bars, Fig. 16d). To acknowledge other layers that are modeled unstable at the same day, we also show the proportion unstable of *all* persistent layers at each day (gray bars, Fig. 16d).

A dashboard view like this effectively summarizes the large amount of simulated observations and allows forecasters to easily put their assessments of specific layers in relation to the simulated data. For example, the reported likelihood of avalanches of persistent avalanche problems shows less variation than the corresponding proportion of simulated unstable grid points (Fig. 16c, d). Focusing on the period from Dec 03 to Jan 05, both bar charts show an almost identical progression. However, while the modeled proportion unstable spans the entire scale from 0–1, the reported likelihood of avalanches only varies between *Possible–Likely/Very Likely*. Simultaneously, the reported danger level is *Low* at first (when the proportion unstable is close to 0), then increases to *High* within three days (Dec 13) (when the proportion unstable also increases to almost 1 at the same time). The danger rating remains constant at *Considerable/High* for roughly one week (when the proportion unstable also lingers between 0.8–1). Starting at Dec 21 the danger rating drops to *Moderate* then *Low* for a total of six days (when the

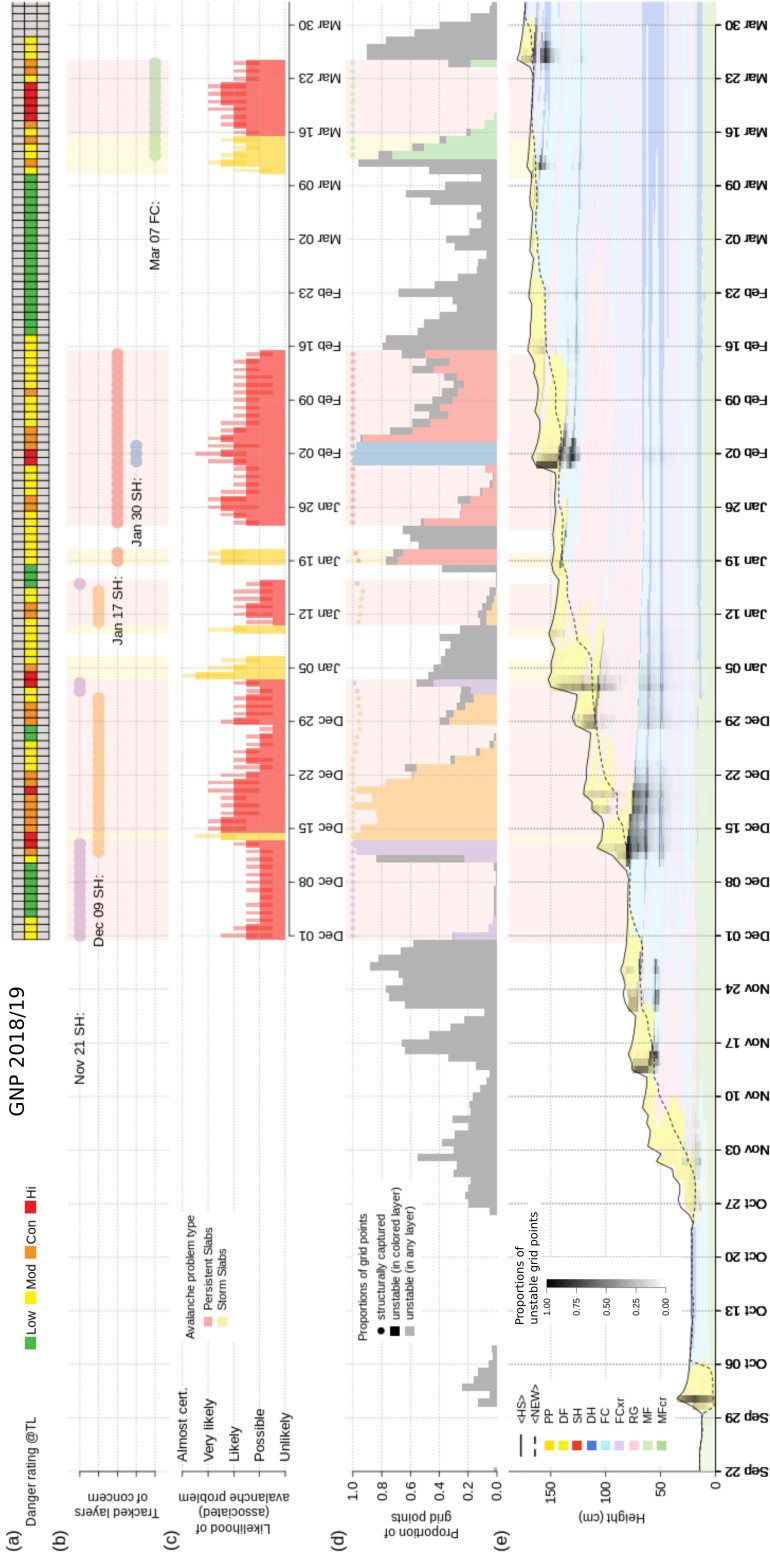

**Figure 16.** A dashboard view that provides a seasonal overview of (a), (b), (c) human assessments and (d), (e) model simulations. (a) The danger rating at treeline elevation (TL), (b) times when tracked layers of concern were associated with an avalanche problem, (c) the likelihood of associated avalanche problems, (d) the proportion of unstable model grid points in the forecast region, and (e) the time series of a representative simulated snow profile, where <HS> and <NEW> refer to the median height of snow and the median amount of new snow, respectively. Acronyms of snow grain types are defined in Sect. 2.2.

proportion unstable also tapers off to 0 within the same time frame) before two snowfall events at Dec 29 and Jan 03 bring the danger rating back up to *Considerable* and *High*. The last two peaks of the danger rating are reflected in the proportion unstable at the same times, but the magnitude remains lower. This aligns with the human assessments that dropped the persistent avalanche problem at Jan 03 and only called it a storm snow problem. At the same time, the average profile (Fig. 16e) highlights

substantial amounts of unstable new snow. Interestingly, it also shows a thin layer of unstable facets that got buried at Dec 28. This layer is not mentioned in the human hazard assessments that still attribute the persistent problem to the Nov 21 SH and the Dec 09 SH layers. Both of these layers are also present in the average profile, but their main activity was modeled between Dec 10 and Dec 23. Besides these nuanced comparisons at times of agreement between modeled and human data sets, the dashboard view makes any serious discrepancies easy to spot. For example, during the early season when no human assessment data is

available yet, the proportion unstable highlights times of instability in early season weak layers, such as around Nov 17 and Nov 24, which are caused by the Nov 14 and Nov 21 layers (Fig. 16d, e). The opposite is possible as well. Starting with Mar 16, the assessments indicate *High* hazard and a persistent avalanche problem on the Mar 07 FC layer. However, the simulations show no signs of instability at all. Despite this dramatic discrepancy, the visualized information can still be beneficial for forecasters who get prompted to think critically about the current situation. Investigating the underlying reasons for the disagreement may

help them make a more informed decision. In this specific example, additional hazard and weather information (not shown in Fig. 16) uncovered that this situation coincided with the first wet avalanche cycle of the year, a process not captured by the stability measure used in this paper.

The second dashboard we envision would focus on a single layer of concern (Fig. 17). Analogously to before, the view combines human assessment information with relevant simulated information. Relevant assessment information could include the

710 times of concern and the likelihood of triggering avalanches due to the associated avalanche problems. To provide meaningful context for the information specific to the layer, the view still includes the proportion captured and the proportion unstable of the specific layer as well as the overall proportion unstable of all persistent layers analogously to the previous view (Fig. 17c, black line with circles, black bars, gray bars, respectively). In addition, the view shows daily summaries of the simulated characteristics of the specific layer of concern, including the layer depth, the statistical stability measure $p_{\text{unstable}}$ (Mayer et al.,

2022), the process-based stability measures RTA, SK38, and $r_c$, and other select variables of interest, such as the cohesion of the overlying slab expressed by the average density over grain size of all slab layers $< \frac{\text{density}}{\text{grain size}} >_{\text{slab}}$. These characteristics are visualized as daily distributions in the form of violin plots. To prevent loss of context in this highly zoomed in view and allow forecasters to relate to information of other potential layers of concern, we also show persistent unstable layers that are *not* associated with the specific layer of concern. These other layers are visualized as separate point clouds color-coded accord-

ing to their native grain types. The novelty of this view is that it selects conceptually equivalent layers based on the methods described in this paper and then visualizes all individual data points from multiple grid points and dates in an accessible way.

The dashboard views shown in Figures 16 and 17 focus on the information contained in the data sets of our present study. In an operational context, additional information will be available that is highly relevant for the proposed validation suite. In particular, adding specific observations like snowfall, snow depth, or avalanche observations will be critical for providing

forecasters with the full range of available evidence. For example, since the proportion unstable will have to be interpreted

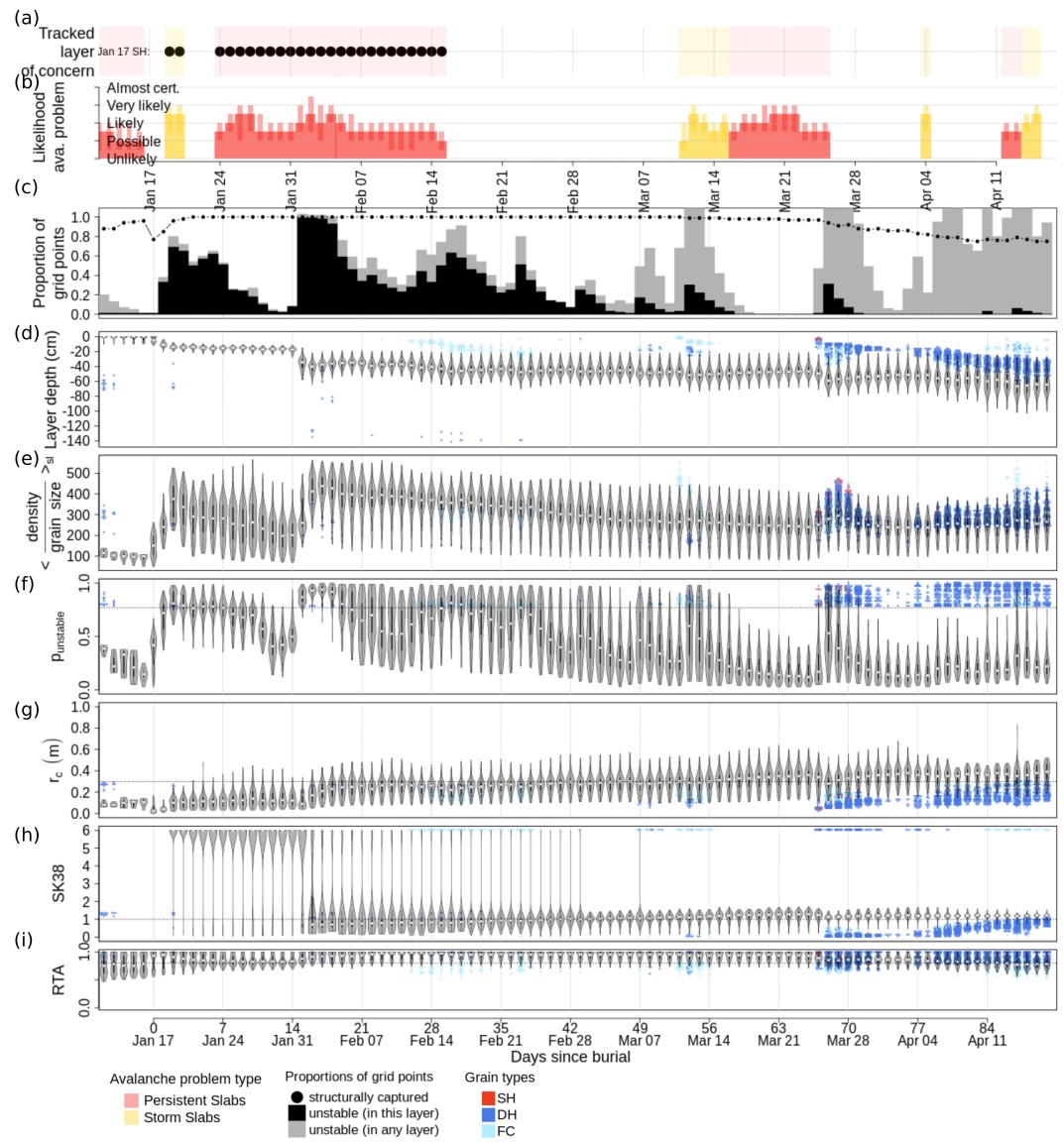

**Figure 17.** A dashboard view that zooms into the details of an individual layer of concern. (a) Times when the layer was associated with an avalanche problem, (b) the likelihood of associated avalanche problems, (c) the proportion of unstable model grid points in the forecast region, and daily summaries of simulated characteristics of the target layer (violin distributions) and other layers (point clouds) like (d) layer depth (m), (e) cohesion of the slab $< \frac{\text{density}}{\text{grain size}} >_{\text{slab}}$ (kg m$^{-3}$ mm$^{-1}$), (f) statistical approach to layer instability $p_{\text{unstable}}$, (g) critical crack length $r_c$ (m), (h) stability index SK38, (i) relative threshold sum approach RTA.

in light of new loading due to added snowfall as currently indicated only in the representative profile, the view could be complemented with observed and modeled snowfall amounts to alert forecasters to situations when modeled snowfall might be substantially under-/overestimated or delayed.

### 5.4 Implications for further development of snowpack modeling

While improving insight for further development of snowpack modeling was not the main objective of this paper, our explorations offer a few interesting observations for this community.

Our large scale persistent layer validation offered an opportunity to compare the overall performance of two recently promoted approaches to estimate dry snow layer instability: The process-based approach promoted by Monti and Schweizer (2013); Monti et al. (2014); Reuter et al. (2015); Monti et al. (2016); Reuter and Schweizer (2018); Reuter et al. (2021) and a statistical approach developed by Mayer et al. (2022). Our results from using the process-based approach are in line with recent research that suggested decreasing the threshold for the critical crack length $r_c$ to 0.3 m (Reuter and Schweizer, 2018; Reuter et al., 2021; Mayer et al., 2022). Furthermore, although the statistical approach was fitted with Swiss data, our results demonstrate that it can be employed in western Canada, where it performs at a similar level as the process-based approach. The proportion of unstable grid points in a forecast region based on the statistical approach turned out to be a valuable predictor of instability. Future research might want to investigate whether a combination of the proportion of unstable grid points with the individual magnitude of instability at each grid point yields improvements.

A further side observation outside of the scope of this paper but nevertheless interesting for developers of snowpack models was brought up by the length of the dry period before the burial of the potential weak layer, which emerged as a variable with strong explanatory power in many of our CTree analyses (Sect. 3.2 and 3.3). Since longer dry periods conceptually increase the chance of weak layer formation and growth on the snow surface, these findings seem plausible at first. However, in sum all of the following observations appear slightly odd: (1) If the dry period before the burial of a SH layer was shorter than 7 days, the proportion of grid points that structurally contained the layer was significantly lower than for longer dry periods (Fig. 12, Nodes 6–8). (2) Barely any SH layer was modeled unstable when buried after a dry period shorter than 7 days, whereas significantly more SH layers were modeled unstable when the dry period was longer (Fig. 13, Nodes 7–9). A less strong but similar effect was observed for FC and combinations of SH/FC (Fig. 13, Nodes 2, 3, 6). (3) Weak layers that were buried after a dry period longer than 12 days were simulated unstable for significantly longer duration than human assessments suggest than weak layers that were buried after a shorter dry period (Fig. 14d). Based on this strong influence of the variable in several different CTree analyses we hypothesize that the model is biased towards overestimating the structural prevalence and instability of weak layers that were buried at the end of long dry periods. Since both structural existence and instability show the same patterns, we further believe that this effect originates in the SNOWPACK model itself and propagates through the stability module by Mayer et al. (2022). To dig even deeper, the variables in the stability module that are potentially affected by the length of the dry period are grain size, density, and sphericity of the weak layer (Mayer et al., 2022). Interestingly, the median simulated grain size at the time of burial was substantially less impactful in our CTree analyses. Hence, we hypothesize that the suggested bias is caused by density or sphericity.

Our data set of human assessments reported multiple SH layers with grain sizes beyond 10 mm, while the median simulated grain sizes at the time of burial rarely exceeded 2.5 mm. Although grain sizes were simulated substantially smaller than reported in many cases, our findings still suggest that the simulated grain sizes were consistent within the simulations. Our CTree analysis has shown that median simulated grain sizes at the time of burial that exceeded 0.9 mm were associated with a higher proportion unstable than smaller grain sizes. This is in line with Mayer et al. (2022), who report an increased influence of the simulated grain size on layer instability for grain sizes above 1–1.5 mm (Fig. B2d in Mayer et al., 2022).

## 6 Conclusion

We evaluated the performance of an operational weather and snowpack model chain to represent persistent weak layers and crusts based on human hazard assessments from three avalanche forecast regions in western Canada and ten winter seasons. By developing methods to identify layers of human concern in the simulations and to group simulated critical layers from different times and grid points by their time of burial, we could quantify (i) the probability that a layer of human concern was captured by the simulations (probability of detection), (ii) the probability that a simulated critical layer was indeed of concern in reality (precision), and (3) the probability that the model simulated a critical layer that is not of concern (false alarm rate). Furthermore, we employed conditional inference trees (CTrees) to (a) identify patterns between layers of concern that were well captured from those that were not and (b) examine the agreement between simulated and reported layers in more detail taking into account the variation and timing of instability.

While we presented model performance curves that will allow forecasters to interpret any model scenario at their hand, the overall model performance can be summarized with a sensitivity of 75 % at a precision of 40 % and a false alarm rate of 30 % when 20 % of model grid points suggest instability in a specific weak layer. While the performance was substantially better for surface hoar (SH) layers than facets (FC), the model had no skill in representing any type of crust layer (IFrc, IFsc, MFcr), neither structurally nor when taking adjacent unstable facets into account. The CTree analyses confirmed the strong influence of grain type. While the analysis of agreement indicators between reported and simulated instability approached the limits of our data sets, it suggested that the simulations and human assessments tended to agree for the majority of captured weak layers.

The presented research contributed to address a knowledge gap in snowpack validation, which has primarily focused on process-based model validation at the point scale and validation of bulk properties at the regional scale. By evaluating the detailed hazardous layering at the regional scale in a way that is informative for forecasters, this contribution will help make snowpack simulations more transparent and applicable for operational avalanche forecasting. Future research may benefit from a more in-depth analysis of the temporal agreement between modeled and reported instability.

We acknowledge that our human validation data set does not represent an objective truth, and the results therefore represent a lower limit of expected model performance. Nevertheless, we believe that the models offer a valuable complementary information source for avalanche forecasting, but we discourage the standalone use of the simulations without any field observations. The methods presented in this paper aim to provide a starting point for designing informative simulation dashboards that enable forecasters to better understand this novel information source by comparing assessments and simulations in real time.

*Code and data availability.* The data and code to reproduce the analysis in this paper are available from a DOI repository at https://www. doi.org/10.17605/OSF.IO/W7PJY (Herla et al., 2023).

**Appendix A: Rules and thresholds to identify model-derived date tags**

We used modeled precipitation amounts to identify model-derived date tags (i.e., potential layers of concern). To describe the precipitation in a study region, we aggregated all solid and liquid precipitation from all model grid points accumulated over 24 and 72 hours by calculating their 25th and 75th percentiles ($<HN24>_{75}$, $<HN72>_{25}$, $<RAIN24>_{75}$, and $<RAIN72>_{25}$). Days with less than a trace amount of precipitation were identified by 24-hour amounts of the 75th percentiles

$<HN24>_{75} < 2$ cm and

$<RAIN24>_{75} < 2$ mm.

We refer to these days as 'Days 0'. To determine which of these Days 0 were followed by a major storm, we scanned the five subsequent days 'Days 1–5' for substantial 72-hour precipitation amounts using the the 25th percentiles and following thresholds

$<HN72>_{25}(@Days 1–5) > 10$ cm or

$<RAIN72>_{25}(@Days 1–5) > 5$ mm.

Whenever any day of Days 1–5 satisfied the above rule, we counted a major storm event. While the above rules recognize the onset of major storms, they can also accidentally identify situations when a major storm tapered off. To exclude these times of tapering-off, we added another set of rules to ensure that the 72-hour precipitation amounts were increasing (i.e., higher than 810 on Day 0) and not decreasing:

$<HN72>_{25}(@Days 1–5) > <HN72>_{25}(@Day 0) + 5$ cm or

$<RAIN72>_{25}(@Days 1–5) > <RAIN72>_{25}(@Day 0) + 2$ mm.

All Days 0 that satisfied the above rules were recorded as model-derived date tags ahead of major storms. To account for long periods in between these main storm events that could have buried persistent weak layers by substantial cumulative accumula-815 tions from small daily snowfall amounts, we inserted additional date tags when a period longer than 10 days accumulated more than 25 cm of $<HN24>_{75}$.

**Appendix B: Tables of acronyms and variables**

This appendix provides a table of acronyms (Table A1) and a table of variables (Table A2) for abbreviations and symbols used throughout the manuscript.

| Acronym | Description |
|---|---|
| ALP | Alpine elevation band |
| BTL | Below treeline elevation band |
| BYK | Banff-Yoho-Kooteney National Park |
| CTree | Conditional inference tree |
| DF | Decomposing and fragmented particles |
| DH | Depth hoar |
| FC | Faceted crystals |
| FCxr | rounding facteted particles |
| $F_1$ | $F_1$ skill score |
| FN | False negative result |
| FP | False positive result |
| GNP | Glacier National Park |
| HRDPS | High resolution deterministic prediction system, a numerical weather prediction model |
| IFrc | Rain crust |
| IFsc | Sun crust |
| MF | Melt forms |
| MFcr | Melt–freeze crust |
| nWKL | Number of weak layers |
| PSS | Peirce skill score |
| PP | Precipitation particles |
| RG | Rounded grains |
| ROC | Receiver operating characteristics curve |
| RTA | Relative threshold sum approach (snow stability index) |
| S2S | Sea-to-Sky avalanche forecast region |
| SH | Surface hoar |
| SK38 | Skier stability index |
| TL | Treeline elevation band |
| TN | True negative result |
| TP | True positive result |

**Table A1.** Descriptions of acronyms used throughout the manuscript.

| Variable | Name (unit) | Description |
|---|---|---|
| $\Delta_{\text{duration}}$ | Difference (days) | Agreement indicator of layer instability |
| $<\frac{\text{density}}{\text{grain size}}>_{\text{slab}}$ | slab cohesion (kg m$^{-3}$ mm$^{-1}$) | Average ratio of density over grain size of the slab |
| $<\text{HN24}>_{75}$ | Solid precipitation (m) | 75th percentile of 24 hour new snow amounts |
| $<\text{HN72}>_{25}$ | Solid precipitation (m) | 25th percentile of 72 hour new snow amounts |
| $\Lambda_{\text{onset}}$ | Lag (days) | Agreement indicator of layer instability |
| $\Lambda_{\text{turn-off}}$ | Lag (days) | Agreement indicator of layer instability |
| $p_{\text{unstable}}$ | Probability of layer instability (1) | A snow stability index |
| $<\text{RAIN24}>_{75}$ | Liquid precipitation (m) | 75th percentile of 24 hour rain amounts |
| $<\text{RAIN72}>_{25}$ | Liquid precipitation (m) | 25th percentile of 72 hour rain amounts |
| $r_c$ | Critical crack length (m) | A mechanical snow layer property |
| $\rho_\Psi$ | Spearman rank correlation (1) | Agreement indicator of layer instability |

**Table A2.** Names, units, and description of variables used throughout the manuscript.

*Author contributions.* All authors conceptualized the research; FH ran the snowpack simulations and implemented the methods and analysis; all authors contributed to writing the paper; PH acquired the funding.

*Competing interests.* FH, SH, and PM declare they have no competing interests; PH is a member of the editorial board of the journal.

*Acknowledgements.* We thank Karl Klassen for valuable feedback and insight throughout the project development.

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
