# Peer review of "A Large-scale Validation of Snowpack Simulations in Support of Avalanche Forecasting Focusing on Critical Layers"

_EGUsphere, 2023_

## Referee Comment (RC1)

**General comments**

The manuscript by Herla et al. reports a validation study. It evaluates the performance of operational snowpack simulations to represent critical layers in view of avalanche triggering at the regional scale. The validation data are qualitative descriptions of potentially relevant layers by avalanche forecasters. The authors acknowledge that those do not represent the truth. Moreover, it seems that these are rather special data and not commonly used in other forecasting services. In general, the impression is that the study is much focused on the Canadian situation and the specific needs of the Canadian forecasters who seem to be skeptical about the usefulness of models. This said, I agree that the comparison presented (regional assessments by forecasters vs. snowpack simulations) is meaningful.

The study is well designed, the methods very appropriate and overall clearly described. Still, I found some paragraphs hard to follow due to some very detailed evaluations that I occasionally had troubles seeing their relevance. Finally, I would personally prefer a slightly stronger emphasis on the research questions and the resulting answers in terms of the application of the model (and future developments) rather than, it seems, directing the discussion primarily to avalanche practitioners.

I recommend the manuscript to be accepted after minor revisions. See my detailed comments below.

**Detailed comments**

Line 28:        I argue that crusts are not critical layers. While persistent weak layers may form above or below crusts, crusts themselves are not critical layers. This misconception should definitely not be further perpetuated.

Lines 53-54:    Please be aware that there are different setups for operational modeling. The most promising is certainly a combination of distributed modeling with continuous data assimilation. In addition, I disagree that simulations at the point scale are not useful. Obviously, almost all measurements and observations are at the point scale, and very often considered useful. As mentioned above, those measurements should then be assimilated in a distributed model.

Line 66:        I wonder whether the term likelihood should only be used in the corresponding statistical context rather than as a synonym for the more generic probability.

Line 94:        Treeline seems to be a rather straightforward to determine objective elevation (range). I wonder why to rely on forecaster consensus.

Line 102:       What do you mean with the "stability scheme" by Michlmayr et al. (2008).

Line 104:       Please provide a short explanation why you turned off SNOWPACK's layer aggregation feature.

Lines 107-110:  I suggest using the official terms for the grain types according to the ICSSG (Fierz et al., 2009) such as "decomposing and fragmented precipitation particles" or "rounded grains".

Line 116:       What is likelihood and size referring to?

Line 184:       Viallon-Galinier et al. (2022) does not seem to be the most appropriate reference to refer to the two main processes of avalanche release.

In another context, the previous snow cover and snow instability modeling study by Reuter and Bellaire (2018) may also be of interest.

Line 186:         Please provide references for the thresholds you selected (RTA, SK38 and $r_c$).

Line 208:         With regard to your note that SH is transformed into DH I wonder since I think to remember that I have also seen SNOWPACK runs where SH was present for longer time periods than just a few days.

Line 227:         snowfall

Figure 3:         I do not follow what you describe in the second last sentence of the caption. Please clarify. Also, to improve readability, I recommend rotating labels on y-axis 90 degrees clockwise, in all figures.

Lines 253-254:    Please clarify: "not assigned a simulated layer"

Line 265:         I suggest using colors that can be better discerned than red and black (vertical lines in Figure 4) or at least use different line style.

Line 266:         delete "in"

Line 270:         Four layers (considered critical by the model) in the text, five in the confusion matrix in Fig. 4d?

Line 330:         delete "can be"

Line 384:         Wilcoxon

Line 399:         Your refer to Fig. 7a,c for the process-based approach, correct?

Figure 9:         The sequence of the subpanels (grain type, elevation, region) is different form the description in the caption. Also, I suggest labeling the curves in panels b, d, f (as in a, c, e).

Line 500:         I suppose not only the performance of the models, but also the forecasts are far from perfect. I guess some of the poor agreement you describe on pages 22-23 may also be related to peculiarities of forecast procedures.

Line 511:         I suppose you refer to *Figure* 13a, here, also below, in line 514.

Line 545:         I agree. On the other hand, you may also consider running slope simulations so that crusts will form on sunny aspects. However, as pointed out, the relevance of crusts compared to weak layers is very different.

Line 550:         Alternatively, apart from snow climate, also forecast performance could be different, or the type of analysis you selected, following so-called layers of concern, may be better suited for GNP.

Line 595:         Likelihood of problem or of avalanches?

Section 5.3:      I recommend you provide some specific examples in Figures 14 and 15 when, e.g. "the view will alert forecasters ", or when the forecast is not supported by model result. For instance, there seems to be little variation in likelihood of avalanches but more in the proportion of unstable grid points – and how does that relate to the danger level?

Figure 14: I suggest adding a legend for the danger rating. Also, please add year and region to the seasonal overview.

Figure 15: There seems to be an error with regard to the units of the slab cohesion. The values you indicate are in the range of 100 to 500, the parameter is supposed to be density divided by grain size, and the units indicated are kg m$^{-4}$?

Lines 674-682: I do not really follow the argumentation here, for instance, why there should be a model bias and what do you refer to by Fig. 12d, 11, and 10? What is the reasoning for density and sphericity for the sensitivity to the length of the dry spell?

Line 691: I suggest replacing human.

Line 701: Is the lack of skill with regard to crusts due to the model setup (flat) or the misconception about the role of crusts?

Line 707: I agree that it is essential to find the critical weak layers and assess their degree of instability. In addition, we may also ask whether the temporal evolution of the instability is properly modeled. In other words, are the parameterizations implemented capable to adequately simulate the temporal evolution of strength and toughness.

Davos, 5 May 2022
Jürg Schweizer

*References*

Reuter, B., and Bellaire, S.: On combining snow cover and snow instability modelling, Proceedings ISSW 2018. International Snow Science Workshop, Innsbruck, Austria, 7-12 October 2018, 949-953, 2018.

Viallon-Galinier, L., Hagenmuller, P., Reuter, B., and Eckert, N.: Modelling snowpack stability from simulated snow stratigraphy: Summary and implementation examples, Cold Reg. Sci. Technol., 201, 103596, https://doi.org/10.1016/j.coldregions.2022.103596, 2022.

---

## Author Comment (AC1)

Discussion of "A Large-scale Validation of Snowpack Simulations in Support of Avalanche Forecasting Focusing on Critical Layers"

**AUTHOR RESPONSE TO REFEREE COMMENT 1**

Herla et al.

June 16, 2023

**1 Responses to Referee #1 (Jürg Schweizer)**

**1.1 General Comments**

Referee Comment: *The manuscript by Herla et al. reports a validation study. It evaluates the performance of operational snowpack simulations to represent critical layers in view of avalanche triggering at the regional scale. The validation data are qualitative descriptions of potentially relevant layers by avalanche forecasters. The authors acknowledge that those do not represent the truth. Moreover, it seems that these are rather special data and not commonly used in other forecasting services. In general, the impression is that the study is much focused on the Canadian situation and the specific needs of the Canadian forecasters who seem to be skeptical about the usefulness of models. This said, I agree that the comparison presented (regional assessments by forecasters vs. snowpack simulations) is meaningful.*

*The study is well designed, the methods very appropriate and overall clearly described. Still, I found some paragraphs hard to follow due to some very detailed evaluations that I occasionally had troubles seeing their relevance. Finally, I would personally prefer a slightly stronger emphasis on the research questions and the resulting answers in terms of the application of the model (and future developments) rather than, it seems, directing the discussion primarily to avalanche practitioners.*

*I recommend the manuscript to be accepted after minor revisions. See my detailed comments below.*

**Author Comment**: Thank you for your positive and encouraging feedback! We value and appreciate your suggestions that have helped us improve the manuscript substantially.

Most comments were editorial, and we think that the revised presentation of the material will help readers better understand the manuscript. We are particularly thankful for your comment 1.2.32, which helped us find an oversight in the performance computation of crust layers. The revision motivated by this non-editorial comment led to a revised methodology in computing the performance curves related to crusts. Due to this revision, the corresponding results improved slightly.

We respond to each comment in a point-by-point manner below.

**1.2 Detailed comments**

**1.2.1 Line 28: Terminology of critical layers**

Referee Comment: *I argue that crusts are not critical layers. While persistent weak layers may form above or below crusts, crusts themselves are not critical layers. This misconception should definitely not be further perpetuated.*

**Author Response:** We thank you for this perspective, which highlights the need for a more in-depth discussion of this topic within our international community. In fact, in Canada the term critical layers that refers to persistent weak layers and crusts has been coined by Bruce Jamieson and his students a while ago. As you describe yourself, persistent weak layers may form adjacent to crusts due to processes that are enhanced by the crusts. The crust layers therefore act as the main reference for practitioners who need to (i) identify these layers in their snow pits, and (ii) extrapolate where these layers might exist and be of concern. The term critical layers (such as used in our first draft manuscript) is based on the incentive to create a concise term. We value both perspectives, but do not think that our manuscript is the right place for this discussion. We therefore accept your comment and changed the use of the term critical layers in our revised manuscript as follows.

Instead of using the term critical layer to *refer* to persistent weak layers and crusts, we use the explicit long version "persistent weak layers and crusts", or substitute the terms "persistent layers", "relevant layers", "layers of concern", and "regional layer" depending on the context. We do keep the term "critical layers" for all layers that are critical with respect to their instability. This is in line with the suggestion by the referee and with the terminology defined in the flowchart of Fig. 2.

**1.2.2 Line 53-54: Limitations of validation studies with respect to the modeling setup**

Referee Comment: *Please be aware that there are different setups for operational modeling. The most promising is certainly a combination of distributed modeling with continuous data assimilation. In addition, I disagree that simulations at the point scale are not useful. Obviously, almost all measurements and observations are at the point scale, and very often considered useful. As mentioned above, those measurements should then be assimilated in a distributed model.*

**Author Response:** Thanks for the comment. We modified the relevant paragraph to be more specific about the limitations we see in existing approaches of these kinds of validation studies. The new edits make the paragraph more clear in

- that we purely refer to validation studies, not the general model setup

- that we do not judge any model setup, and

- that the magnitude of the limitations depends on the region of interest and the setup of the operational model.

The revised paragraph reads as follows:

> "Despite this large body of snowpack validation studies, the operational needs of (Canadian) avalanche forecasters have not been satisfied yet. While process-based validations **at individual point locations** based on high quality data are crucial for model development and improvement, **these validation results** do not provide **sufficiently** tangible and relevant guidance for forecasters **who forecast for different locations or regions. In addition, these validation results** are not necessarily representative of the real skill of operational simulations **which might rely on different data sources or model configurations.** [...] "

**1.2.3 Line 66: Terminology likelihood vs probability**

Referee Comment: *I wonder whether the term likelihood should only be used in the corresponding statistical context rather than as a synonym for the more generic probability.*

**Author Response:** We agree, thanks for pointing this out! We changed all occurrences of the term "likelihood" to "probability" where applicable. Please note, that we kept the term "likelihood" unchanged where we refer to the likelihood component of an avalanche problem as defined in the Conceptual Model of Avalanche Hazard (Statham et al. 2018).

**1.2.4 Line 94: Treeline elevation range**

Referee Comment: *Treeline seems to be a rather straightforward to determine objective elevation (range). I wonder why to rely on forecaster consensus.*

**Author Comment:** Forecaster consensus was chosen for treeline elevation ranges because operational forecasters do not have consistent objective definitions for vegetation bands. We wanted to know their interpretations of elevation ranges in each region because they would be matching field observations from those ranges to vegetation bands in their assessments. This should create the best possible match in elevation between assessments and profile simulations.

As an aside, work towards an objective classification of vegetation bands in western Canada (where treeline ranges from 800 to 2400 m) has been challenging because forest density data tends to be poor quality at upper elevations and it is difficult to detect transitions in areas without alpine terrain.

The slightly modified paragraph reads as follows:

> We classified each grid point into an elevation band class, 'alpine' (ALP), 'treeline' (TL), and 'below treeline' (BTL) to match the terrain classification in the human assessments. To create the best possible match in elevation between assessments and simulations, we used forecaster consensus of TL elevation for the classification: 1600–1800 m in S2S, 1800–2100 m in GNP, and 2000–2400 m in BYK.

**1.2.5 Line 102: Stability scheme**

Referee Comment: *What do you mean with the "stability scheme" by Michlmayr et al. (2008).*

**Author Response:** We refer to the *atmospheric* stability scheme. We make that clear in the revised version.

**1.2.6 Line 104: SNOWPACK's layer aggregation**

Referee Comment: *Please provide a short explanation why you turned off SNOWPACK's layer aggregation feature.*

**Author Response:** The newly added explanation reads as follows:

> Lastly, we turned off SNOWPACK's layer aggregation feature of merging similar and adjacent layers to preserve exact knowledge of the formation dates of individual layers. The need for this decision will become apparent in Sect. 3.1.2 and 3.1.3, where we explain our approach of grouping and matching layers based on date considerations.

**1.2.7 Line 107-110: Official grain type terminology**

Referee Comment: *I suggest using the official terms for the grain types according to the ICSSG (Fierz et al., 2009) such as "decomposing and fragmented precipitation particles" or "rounded grains".*

**Author Response:** Agreed. Changed accordingly.

**1.2.8 Line 116: Definition of likelihood and size**

Referee Comment: *What is likelihood and size referring to?*

**Author Response:** In the revised version, we explain in more detail:

Applying the Conceptual Model of Avalanche Hazard (Statham et al. 2018), forecasters partitioned the avalanche hazard into different avalanche problems and characterized each problem by its type, location, likelihood of avalanches, and destructive avalanche size.

**1.2.9 Line 184: Change reference**

Referee Comment: *Viallon-Galinier et al. (2022) does not seem to be the most appropriate reference to refer to the two main processes of avalanche release.*

*In another context, the previous snow cover and snow instability modeling study by Reuter and Bellaire (2018) may also be of interest.*

**Author Response:** We changed the previous reference (most recent review of stability indices in snowpack models) to more general reviews on slab avalanche formation.

[...] which accounts for the two main processes governing slab avalanche release (Schweizer et al. 2003, 2016).

We extensively refer to Reuter et al. (2021), which is the peer-reviewed development of Reuter and Bellaire (2018). We added Reuter and Bellaire (2018) to the revised Introduction.

**1.2.10 Line 186: Add references**

Referee Comment: *Please provide references for the thresholds you selected (RTA, SK38 and $r_c$ ).*

**Author Response:** Agreed. The revised version reads as follows:

[...] While the literature agrees on thresholds for RTA and SK38, less consensus exists for $r_c$. Hence, we applied two thresholds to $r_c$ and therefore identified critical avalanche layers with poor stability if RTA $\geq 0.8$ (Monti and Schweizer 2013), SK38 $\leq 1$ (Reuter et al. 2021), and $r_c \leq 0.3$ (Reuter et al. 2021) or $r_c \leq 0.4$ (Reuter et al. 2015).

**1.2.11 Line 208: SNOWPACK transforming DH-SH**

Referee Comment: *With regard to your note that SH is transformed into DH I wonder since I think to remember that I have also seen SNOWPACK runs where SH was present for longer time periods than just a few days.*

**Author Response:** We have seen the full range from a few days to several weeks. In the revised manuscript, we change the wording to also account for those cases where SH is present for longer.

> [...] To address the well-known SNOWPACK behaviour of transforming most SH layers into DH layers after they have been buried for several days to weeks, [...]

**1.2.12 Line 227: typo**

Referee Comment: *snowfall*

**Author Response:** Changed. Thank you!

**1.2.13 Figure 3: Caption and y-axis labels**

Referee Comment: *I do not follow what you describe in the second last sentence of the caption. Please clarify. Also, to improve readability, I recommend rotating labels on y-axis 90 degrees clockwise, in all figures.*

**Author Response:** Thanks for the heads up! We changed the caption of Figure 3 and will rotate the y-axis labels in all figures. The updated Figure 3 with its revised caption:

[Figure]

An illustration of the search windows for layers of human concern around the human date tag. The panel zooms in on the near-surface layers of a time series of a simulated snow profile. The extent of the formation and burial windows are highlighted by horizontal lines, where the solid lines indicate the extent of each window based on the timing of the storms. The black boxes highlight all layers that either formed within the formation window or got buried within the burial window. To better illustrate the effect that different lengths of time windows have on the selection of layers, the gray hatched lines highlight layers that would result from fixed time windows that are not based on the timing of the storms (dashed horizontal lines). Colors refer to snow grain types defined in Sect. 2.2.

**1.2.14 Line 253-254: Assigning simulated layers to datetags**

Referee Comment: *Please clarify: "not assigned a simulated layer"*

**Author Response:** Thanks for the comment that this concept is difficult to follow. We simplified the wording and inserted a reference to the section where this concept is explained in detail. The revised sentence reads as follows:

> Analogously, all potential model-derived date tags without any simulated unstable layers (rightmost arrow in Fig. 2, Sect. 3.1.2) were counted as absent layers and directly contributed to the True Negative cell of the matrix.

**1.2.15 Line 265/Figure 4: Different line colors**

Referee Comment: *I suggest using colors that can be better discerned than red and black (vertical lines in Figure 4) or at least use different line style.*

**Author Response:** Agreed. We will change that accordingly.

**1.2.16 Line 266: Typo**

Referee Comment: *delete "in"*

**Author Response:** Done. Thank you.

**1.2.17 Line 270: Typo**

Referee Comment: *Four layers (considered critical by the model) in the text, five in the confusion matrix in Fig. 4d?*

**Author Response:** Thank you for spotting this! This is a typo in the text. Figure 4a shows that the number 5 in the confusion matrix of Fig. 4d is correct. Changed.

**1.2.18 Line 330: Typo**

Referee Comment: *delete "can be"*

**Author Response:** Done. Thank you.

**1.2.19 Line 384: Typo**

Referee Comment: *Wilcoxon*

**Author Response:** Changed. Thank you.

**1.2.20 Line 384: Figure reference**

Referee Comment: *Your refer to Fig. 7a,c for the process-based approach, correct?*

**Author Response:** Yes, the figure reference (Fig. 7a, c) is correct. We edited the sentence slightly to clarify potential ambiguity:

> The performance of the process-based approach with the higher threshold is significantly lower than the statistical approach on the 95 % confidence level. The process-based approach with the lower threshold, however, yields comparable results to the statistical approach and lies within the confidence band of the statistical approach (Fig. 7a, c).

**1.2.21 Figure 9: Labels**

Referee Comment: *The sequence of the subpanels (grain type, elevation, region) is different form the description in the caption. Also, I suggest labeling the curves in panels b, d, f (as in a, c, e).*

**Author Response:** Good catch! We corrected the description in the caption and added labels to the curves in panels b, d, f.

**1.2.22 Line 500: Performance of modeled and human forecasts**

Referee Comment: *I suppose not only the performance of the models, but also the forecasts are far from perfect. I guess some of the poor agreement you describe on pages 22-23 may also be related to peculiarities of forecast procedures.*

**Author Response:** Absolutely, we agree. To pick up readers with the same thought, we added a sentence about this fact and then refer to Sect. 5.2, where we discuss this in detail. The revised introductory paragraph to Sect. 5.1 reads as follows:

> Our results have shown that the Canadian weather and snowpack model chain is able to skillfully represent weak layers that are of operational concern. However, comparison against human assessments also demonstrates substantial differences between the two data sets. Both data sets are characterized by uncertainty, and both forecasts—modeled and human—are far from perfect. While this fact is discussed in detail in the subsequent section (Sect. 5.2 Limitations), the current section [...].

**1.2.23 Line 511: Typo**

Referee Comment: *I suppose you refer to Figure 13a, here, also below, in line 514.*

**Author Response:** Correct, thank you. Changed.

**1.2.24 Line 545: Recommendation for crusts**

Referee Comment: *I agree. On the other hand, you may also consider running slope simulations so that crusts will form on sunny aspects. However, as pointed out, the relevance of crusts compared to weak layers is very different.*

**Author Response:** Agreed. We added the recommendation.

> Given the poor model performance in capturing any type of crust, while modeling many additional crust layers that were never reported upon, we advise to use the Canadian operational weather and snowpack model chain only for assessing weak layers (such as SH, DH, FC) and not crusts, at least until the model chain includes slope simulations and further testing has been carried out.

**1.2.25 Line 550: Alternative effects influencing performance in different regions**

Referee Comment: *Alternatively, apart from snow climate, also forecast performance could be different, or the type of analysis you selected, following so-called layers of concern, may be better suited for GNP.*

**Author Response:** That is correct. We added two statements in the existing paragraph to clarify this:

This strong effect is likely influenced by the regions' snow climates, **as well as forecast practices by different agencies in different snow climates.** Due to its transitional snow climate GNP is characterized by substantial snowfall amounts interspersed with frequent periods of critical layer formation (Haegeli and McClung 2007; Shandro and Haegeli 2018). Therefore many persistent avalanche problems exist each season, most of which can usually be linked to specific thin critical layers (Fig. 5c, d). In the maritime snow climate of S2S, critical layers form much less often and cause less persistent problems (Fig. 5a, b). Although continental BYK experiences the most days per season with persistent avalanche problems, less persistent problems can actually be linked to specific critical layers (Fig. 5e, f). Instead of thin critical layers, the continental snowpack is often characterized by thick bulk layers of low cohesion. Since these thick layers often get deposited by different snowfall events and facet over the course of many dry spells, it can be challenging to name these layers, let alone distinguish them in the field. **Our analysis approach of focusing on specific identifiable layers of concern may therefore be most applicable to GNP.** For all these reasons our data set of tracked layers of concern is skewed towards GNP and leaves BYK and particularly S2S underrepresented. Our results for S2S and BYK therefore have to be interpreted with more caution and in light of their regional peculiarities. For example, Horton and Haegeli (2022) found that BYK (and other forecast regions in the Canadian Rockies) consistently receive underestimated modeled snowfall amounts, which increases temperature gradients in the snowpack and helps explain the overestimated faceting we see in our results of BYK, which in turn leads to a low model precision in BYK. In contrast, the low model precision in S2S could be due to an underrepresentation of layers of concern. Since instabilities are often short-lived when persistent weak layers get buried by big storms, many critical layers potentially get never associated with a persistent avalanche problem.

**1.2.26 Line 595: Likelihood terminology**

Referee Comment: *Likelihood of problem or of avalanches?*

**Author Response:** Likelihood of avalanches given a specific avalanche problem. We changed the statement accordingly.

**1.2.27 Section 5.3: Ideas for specific examples**

Referee Comment: *I recommend you provide some specific examples in Figures 14 and 15 when, e.g. "the view will alert forecasters", or when the forecast is not supported by model result. For instance, there seems to be little variation in likelihood of avalanches but more in the proportion of unstable grid points – and how does that relate to the danger level?*

**Author Response:** Thanks for this recommendation. We absolutely see the added value of these tangible examples, and added an entire paragraph to the section:

A dashboard view like this effectively summarizes the large amount of simulated observations and allows forecasters to easily put their assessments of specific layers in relation to the simulated data. For example, the reported likelihood of avalanches of persistent avalanche problems shows less variation than the corresponding proportion of simulated unstable grid points (Fig. 14c, d). Focusing on the period from Dec 03 to Jan 05, both bar charts show an almost identical progression. However, while the modeled proportion unstable spans the entire scale from 0–1, the reported likelihood of avalanches only varies between *Possible–Likely/Very Likely*. Simultaneously, the

reported danger level is *Low* at first (when the proportion unstable is close to 0), then increases to *High* within three days (Dec 13) (when the proportion unstable also increases to almost 1 at the same time). The danger rating remains constant at *Considerable/High* for roughly one week (when the proportion unstable also lingers between 0.8–1). Starting at Dec 21 the danger rating drops to *Moderate* then *Low* for a total of six days (when the proportion unstable also tapers off to 0 within the same time frame) before two snowfall events at Dec 29 and Jan 03 bring the danger rating back up to *Considerable* and *High*. The last two peaks of the danger rating are reflected in the proportion unstable at the same times, but the magnitude remains lower. This aligns with the human assessments that dropped the persistent avalanche problem at Jan 03 and only called it a storm snow problem. At the same time, the average profile (Fig. 14e) highlights substantial amounts of unstable new snow. Interestingly, it also shows a thin layer of unstable facets that got buried at Dec 28. This layer is not mentioned in the human hazard assessments that still attribute the persistent problem to the Nov 21 SH and the Dec 09 SH layers. Both of these layers are also present in the average profile, but their main activity was modeled between Dec 10 and Dec 23. Besides these nuanced comparisons at times of agreement between modeled and human data sets, the dashboard view makes any serious discrepancies easy to spot. For example, during the early season when no human assessment data is available yet, the proportion unstable highlights times of instability in early season weak layers, such as around Nov 17 and Nov 24, which are caused by the Nov 14 and Nov 21 layers (Fig. 14d, e). The opposite is possible as well. Starting with Mar 16, the assessments indicate *High* hazard and a persistent avalanche problem on the Mar 07 FC layer. However, the simulations show no signs of instability at all. Despite this dramatic discrepancy, the visualized information can still be beneficial for forecasters who get prompted to think critically about the current situation. Investigating the underlying reasons for the disagreement may help them make a more informed decision. In this specific example, additional hazard and weather information (not shown in Fig. 14) uncovered that this situation coincided with the first wet avalanche cycle of the year, a process not captured by the stability measure used in this paper.

**1.2.28 Figure 14: Figure legend**

Referee Comment: *I suggest adding a legend for the danger rating. Also, please add year and region to the seasonal overview.*

**Author Response:** Agreed. We will add the legend and add the year/season to the figure.

**1.2.29 Figure 15: Unit error**

Referee Comment: *There seems to be an error with regard to the units of the slab cohesion. The values you indicate are in the range of 100 to 500, the parameter is supposed to be density divided by grain size, and the units indicated are kg m -4 ?*

**Author Response:** Thank you for pointing this out. Indeed, the units need to read kg m$^{-3}$ mm$^{-1}$. This choice of units is consistent with Mayer et al. (2022, Fig. B2f). We changed the unit error in the manuscript.

**1.2.30 Lines 674-682: Paragraph needs rewrite**

Referee Comment: *I do not really follow the argumentation here, for instance, why there should be a model bias and what do you refer to by Fig. 12d, 11, and 10? What is the reasoning for*

*density and sphericity for the sensitivity to the length of the dry spell?*

**Author Response:** That comment calls for a thorough revision of the entire paragraph. We added explanations to improve the logic of the idea, stated explicitly that this idea is a hypothesis, and we separated the discussion of grain size from the discussion of the potential bias. The revised version reads as follows:

> A further side observation outside of the scope of this paper but nevertheless interesting for developers of snowpack models was brought up by the length of the dry period before the burial of the potential weak layer, which emerged as a variable with strong explanatory power in many of our CTree analyses (Sect. 3.2 and 3.3). Since longer dry periods conceptually increase the chance of weak layer formation and growth on the snow surface, these findings seem plausible at first. However, in sum all of the following observations appear slightly odd: (1) If the dry period before the burial of a SH layer was shorter than 7 days, the proportion of grid points that structurally contained the layer was significantly lower than for longer dry periods (Fig. 10, Nodes 6–8). (2) Barely any SH layer was modeled unstable when buried after a dry period shorter than 7 days, whereas significantly more SH layers were modeled unstable when the dry period was longer (Fig. 11, Nodes 7–9). A less strong but similar effect was observed for FC and combinations of SH/FC (Fig. 1, Nodes 2, 3, 6). (3) Weak layers that were buried after a dry period longer than 12 days were simulated unstable for significantly longer duration than human assessments suggest than weak layers that were buried after a shorter dry period (Fig. 12d). Based on this strong influence of the variable in several different CTree analyses we hypothesize that the model is biased towards overestimating the structural prevalence and instability of weak layers that were buried at the end of long dry periods. Since both structural existence and instability show the same patterns, we further believe that this effect originates in the SNOWPACK model itself and propagates through the stability module by Mayer et al. (2022). To dig even deeper, the variables in the stability module that are potentially affected by the length of the dry period are grain size, density, and sphericity of the weak layer (Mayer et al. 2022). Interestingly, the median simulated grain size at the time of burial was substantially less impactful in our CTree analyses. Hence, we hypothesize that the suggested bias is caused by density or sphericity.
>
> Our data set of human assessments reported multiple SH layers with grain sizes beyond 10 mm, while the median simulated grain sizes at the time of burial rarely exceeded 2.5 mm. Although grain sizes were simulated substantially smaller than reported in many cases, our findings still suggest that the simulated grain sizes were consistent within the simulations. Our CTree analysis has shown that median simulated grain sizes at the time of burial that exceeded 0.9 mm were associated with a higher proportion unstable than smaller grain sizes. This is in line with Mayer et al. (2022), who report an increased influence of the simulated grain size on layer instability for grain sizes above 1–1.5 mm (Fig. B2d in Mayer et al. 2022).

**1.2.31 Lines 691: Language edit**

Referee Comment: *I suggest replacing human.*

**Author Response:** Agreed. Done.

**1.2.32 Lines 701: Performance of crusts**

Referee Comment: *Is the lack of skill with regard to crusts due to the model setup (flat) or the misconception about the role of crusts?*

**Author Response:** At the outset of this project, we actually planned to solely look into structural presence or absence of weak layers and crusts. As the project matured, we decided to take stability into account as well. While implementing that additional step, we indeed incorporated this oversight. Upon verifying our calculations due to this comment, we found that we only looked at the stability of the crust grains instead of the stability of adjacent facets. That is an error that impacts the results presented in Fig. 9a, b, curve labeled 'MFcr'. In the meantime, we ran the relevant calculations, and we have made the following edits to the manuscript.

We added the appropriate routines for selecting crust layers with adjacent unstable faceted layers:

> To identify relevant layers in the snowpack simulations, we searched all simulated snow profiles for layers that were characterized by persistent grain types and poor stability. Persistent grain types included all faceted layers (FC, DH), surface hoar layers (SH), and crust layers (MFcr, IF). Since crust layers are strong layers, we counted crust layers as unstable if unstable weak layers (FC) were present in the vicinity of the crusts.
>
> [...]
>
> While the search windows identified all simulated layers that align with a specific human date tag, the layers still needed to be filtered for the reported grain types to produce a meaningful match. To accomplish this, we employed up to four different grain type searches for each layer of concern: a) a strict grain type search that only accepted the specific *reported* weak grain types (e.g., SH, SH/FC, etc.) in the simulated layers; b) a relaxed grain type search that accepted *all* persistent weak grain types (SH/DH/FC) in the simulated layers; and a grain type search for c) any crust layer (IF, MFcr) or d) any crust layer with an adjacent unstable faceted layer if crusts were reported to play a role in the given layer of concern.

We re-computed Fig. 9a, b where the curves that represent crusts changed:

[Figure]

We changed the description of the figure in the text:

Crust layers are poorly represented in the simulations with probabilities of detection below 50 % at a low precision of 10-20%. All performance measures for crusts closely follow the no-skill base line (Fig. 9a, b).

We added a statement about the poor precision in the discussion, which was not an issue in the old version:

Given the poor model performance in capturing any type of crust, while modeling many additional crust layers that were never reported upon, we advise to use the Canadian operational weather and snowpack model chain only for assessing weak layers (such as SH, DH, FC) and not crusts, at least until the model chain includes slope simulations and further testing has been carried out.

We added a statement to the concluding comment:

While the performance was substantially better for surface hoar (SH) layers than facets (FC), the model had no skill in representing any type of crust layer (IFrc, IFsc, MFcr), neither structurally nor when taking adjacent unstable facets into account.

Overall, the revised methodology is now in line with the referee's comments, and the crust performance improved slightly from extremely poor to poor. We believe that the reason for the poor performance is related to (i) the configuration of the simulations (flat field instead of slopes) for IFsc layers, and (ii) the phase determination of precipitation based on a temperature threshold implemented in SNOWPACK (instead of using phase information from the weather model) and (iii) other meteorological inputs from the weather model for IFrc and MFcr layers.

**1.2.33 Lines 707: Conclusion**

Referee Comment: *I agree that it is essential to find the critical weak layers and assess their degree of instability. In addition, we may also ask whether the temporal evolution of the instability is properly modeled. In other words, are the parameterizations implemented capable to adequately simulate the temporal evolution of strength and toughness.*

**Author Response:** Thanks for this encouraging comment—this is what we're working on right now ;-). Although again on the regional scale and constrained by our data set of human assessments, so our results will likely not contribute to evaluating the parametrizations you mention. I'm curious whether Switzerland has the data to run these validations on the regional scale (around Davos?) and for many winter seasons?

We added one more sentence to the end of the paragraph:

Future research may benefit from a more in-depth analysis of the temporal agreement between modeled and reported instability.

**References**

Haegeli, P. and McClung, D. M.: Expanding the snow-climate classification with avalanche-relevant information: Initial description of avalanche winter regimes for southwestern Canada, J Glaciol, 53, 266–276, https://doi.org/10.3189/172756507782202801, 2007.

Horton, S. and Haegeli, P.: Using snow depth observations to provide insight into the quality of snowpack simulations for regional-scale avalanche forecasting, The Cryosphere, 16, 3393–3411, https://doi.org/10.5194/tc-16-3393-2022, 2022.

Mayer, S., van Herwijnen, A., Techel, F., and Schweizer, J.: A random forest model to assess snow instability from simulated snow stratigraphy, The Cryosphere, 16, 4593–4615, https://doi.org/10.5194/tc-16-4593-2022, 2022.

Monti, F. and Schweizer, J.: A relative difference approach to detect potential weak layers within a snow profile, in: Proceedings of the 2013 International Snow Science Workshop, Grenoble, France, pp. 339–343, URL `https://arc.lib.montana.edu/snow-science/item.php?id=1861`, 2013.

Reuter, B. and Bellaire, S.: On combining snow cover and snow instability modelling, in: Proceedings of the 2018 international snow science workshop, Innsbruck, AUT, pp. 949—-953, URL `https://arc.lib.montana.edu/snow-science/item/2684`, 2018.

Reuter, B., Schweizer, J., and van Herwijnen, A.: A process-based approach to estimate point snow instability, The Cryosphere, 9, 837–847, 2015.

Reuter, B., Viallon-Galinier, L., Horton, S., van Herwijnen, A., Mayer, S., Hagenmuller, P., and Morin, S.: Characterizing snow instability with avalanche problem types derived from snow cover simulations, Cold Reg Sci Technol, 194, 103 462, https://doi.org/10.1016/j.coldregions.2021.103462, 2021.

Schweizer, J., Jamieson, J. B., and Schneebeli, M.: Snow avalanche formation, Rev Geophys, 41, https://doi.org/10.1029/2002rg000123, 2003.

Schweizer, J., Reuter, B., Van Herwijnen, A., and Gaume, J.: Avalanche Release 101, in: Proceedings of the 2016 International Snow Science Workshop, Breckenridge, CO, USA, URL `https://arc.lib.montana.edu/snow-science/item/2235`, 2016.

Shandro, B. and Haegeli, P.: Characterizing the nature and variability of avalanche hazard in western Canada, Nat Hazard Earth Sys, 18, 1141–1158, https://doi.org/10.5194/nhess-18-1141-2018, 2018.

Statham, G., Haegeli, P., Greene, E., Birkeland, K. W., Israelson, C., Tremper, B., Stethem, C., McMahon, B., White, B., and Kelly, J.: A conceptual model of avalanche hazard, Nat Hazards, 90, 663–691, https://doi.org/10.1007/s11069-017-3070-5, 2018.

---

## Author Comment (AC2)

Discussion of "A Large-scale Validation of Snowpack Simulations in Support of Avalanche Forecasting Focusing on Critical Layers"

**Author response to community comment 1**

Herla et al.

June 14, 2023

**1 Responses to community comment #1 (Ron Simenhois)**

**1.1 General Comments**

**1.1.1 Kudos**

Comment: *This is a well-written and interesting manuscript with impressive results. Below are a few general comments on the manuscript.*

**Author Comment**: Thank you very much for the kudos, and thank you for reaching out with your ideas of improving the manuscript! We really appreciate it!

**1.1.2 Influence of the slab**

Comment: *This work focuses on identifying weak layers for avalanche forecasting. However, a large body of research highlights the importance of both the weak layer and slab for avalanche formation. I believe the manuscript will benefit from a short discussion about why this work focuses on weak layers only.*

**Author Response:** Thanks for this suggestion! As you mention, at any given grid point and any given time, the snowpack instability depends on weak layer and slab characteristics. We actually take both of these characteristics into account when computing the stability of each layer in our data set. Please refer to comment 1.1.4 for a more detailed response and for how we made this more explicit in the revised manuscript.

This said, the general focus of our study is indeed on the weak layer. Since the slab is primarily a function of precipitation (Richter et al. 2020), research that contributes to our understanding of the modeled slab would rather look at simulated snowfall. Horton and Haegeli (2022) did exactly that for the same model chain that this paper uses. We added the following statement to the Introduction to create the mentioned link between snowfall validation research and the slab:

> Regional-scale validation studies of simulated snowfall further contribute essential information to the valuation of snowpack simulations for avalanche forecasting, particularly with respect to snow surface avalanche problems (e.g., storm snow problems) and characteristics of the slab, which is primarily influenced by precipitation (Richter

et al. 2020). Nevertheless, the existing research does not paint a comprehensive picture yet: to our knowledge no large-scale study exists that created a specific link between simulated layers and known critical avalanche layers.

In our analysis, the slab is most impactful in Sect. 4.3.3, where we compare temporal trends of modeled and reported instability. For situations with poor agreement, we cannot disentangle whether the weak layer or the slab is the main culprit. We added a comment to Sect. 5.2 "Limitations", where we discuss this challenge:

> Lastly, any temporal comparisons between simulations and assessments are additionally impacted by uncertainties in snowfall frequency and magnitude, which in turn impact weak layer and slab characteristics alike. Again, it often remains unclear which data source is closer to reality when new snow amounts differ (Lundquist et al. 2019; Horton and Haegeli 2022).

**1.1.3 Dry slab avalanche forecasting**

Comment: *The authors mention that this work aims to help with dry avalanche forecasting toward the end of the manuscript. It will add to the manuscript's clarity if this is mentioned earlier.*

**Author Response:** Agreed. We added statements in the Abstract and Introduction that will clarify this early during the reading.

> These simulations contain information about thin, persistent critical avalanche layers that are buried within the snowpack and are fundamental drivers of **dry slab avalanche hazard.**
>
> [...]
>
> Overall, our study quantifies the capabilities of an operational weather and snowpack model chain to represent critical avalanche layers **that are prone to cause dry slab avalanches,** which contributes to making snowpack simulations more transparent and applicable for operational applications.

**1.1.4 Weak layer characteristics**

Comment: *This work concentrates on snowpack layers' grain type (persistent weak layers). Other weak layer characteristics that may or may not be more important than grain type, like depth, hardness, grain size, etc., are only mentioned toward the end as potential for future work. Again, I think the manuscript will benefit from a short discussion of why these weak layer characteristics are not part of this work.*

**Author Response:** In a first instance, we use grain type to identify layers that will cause persistent avalanche problems and will therefore be of concern substantially longer than non-persistent critical layers (e.g., storm snow). Once we had identified these persistent layers, a variety of weak layer and slab characteristics was indeed used to assess their modeled instability. We added explicit statements to Sect. 3.1.2 (Pre-processing simulations) to clarify this.

> [...] the model learned to predict the probability of layer instability ($p_{unstable}$) from a set of six simulated predictor variables. These predictor variables included characteristics of both the weak layer and the slab, namely the viscous deformation rate, the critical cut length, the sphericity and grain size of the weak layer, the skier penetration depth, and the cohesion of the slab. [...]

[...] our process-based approach used a combination of three indices to assess the instability of a layer: a) the relative threshold sum approach RTA (Monti and Schweizer 2013; Monti et al. 2014), b) the multi-layered skier stability index SK38 (Monti et al. 2016), and c) the critical crack length $r_c$ (Richter et al. 2019). Each of these indices consists of a variety of weak layer and slab characteristics, such as macroscopic properties (e.g., layer depth), microstructural properties (e.g., grain type and size), and mechanical properties (e.g., shear strength). [...]

This said, it is correct, though, that the presentation of our results is very much influenced by grain type. Not by personal choice, but informed by our CTree analyses, where grain type emerged as a strong explanatory variable. Now, you could ask why we didn't include any other weak layer characteristics other than grain type and grain size at the time of burial. We added an explanation to Sect. 3.2, where we explain our choice of explanatory variables:

We included the following potential explanatory variables in our analyses, which contained both direct layer attributes as well as more contextual information:

- grain type (and grain type search routine)
- grain size of the simulated layer at burial
- data quality class
- month of burial
- number of associated avalanche problem days
- simulated length of the dry spell before burial
- season
- region and elevation band.

Since the CTree analysis required the explanatory variables to remain constant over the lifetime of the layer, we did not include any other specific weak layer or slab variables.

To sum up, the added explanations should make it more transparent for the future reader that a variety of weak layer and slab characteristics are indeed considered in our methodology, but that those have not been considered as potential explanatory variables of the CTree analysis.

**References**

Horton, S. and Haegeli, P.: Using snow depth observations to provide insight into the quality of snowpack simulations for regional-scale avalanche forecasting, The Cryosphere, 16, 3393–3411, https://doi.org/10.5194/tc-16-3393-2022, 2022.

Lundquist, J., Hughes, M., Gutmann, E., and Kapnick, S.: Our skill in modeling mountain rain and snow is bypassing the skill of our observational networks, Bull Am Meteorol Soc, 100, 2473–2490, https://doi.org/10.1175/BAMS-D-19-0001.1, 2019.

Monti, F. and Schweizer, J.: A relative difference approach to detect potential weak layers within a snow profile, in: Proceedings of the 2013 International Snow Science Workshop, Grenoble, France, pp. 339–343, URL `https://arc.lib.montana.edu/snow-science/item.php?id=1861`, 2013.

Monti, F., Schweizer, J., Gaume, J., and Fierz, C.: Deriving snow stability information from simulated snow cover stratigraphy, in: Proceedings of the 2014 International Snow Science Workshop, Banff, AB, Canada, pp. 465–469, URL `https://arc.lib.montana.edu/snow-science/item.php?id=2096`, 2014.

Monti, F., Gaume, J., van Herwijnen, A., and Schweizer, J.: Snow instability evaluation: calculating the skier-induced stress in a multi-layered snowpack, Nat Hazard Earth Sys, 16, 775–788, https://doi.org/10.5194/nhess-16-775-2016, 2016.

Richter, B., Schweizer, J., Rotach, M. W., and Van Herwijnen, A.: Validating modeled critical crack length for crack propagation in the snow cover model SNOWPACK, The Cryosphere, 13, 3353–3366, https://doi.org/10.5194/tc-13-3353-2019, 2019.

Richter, B., Van Herwijnen, A., Rotach, M. W., and Schweizer, J.: Sensitivity of modeled snow stability data to meteorological input uncertainty, Nat Hazard Earth Sys, 20, 2873–2888, https://doi.org/10.5194/nhess-20-2873-2020, 2020.

---

## Author Comment (AC3)

Discussion of "A Large-scale Validation of Snowpack Simulations in Support of Avalanche Forecasting Focusing on Critical Layers"

**AUTHOR RESPONSE TO REFEREE COMMENT 1**

Herla et al.

May 8, 2024

**1 Responses to Referee #2 (anonymous)**

**1.1 General Comments**

Referee Comment: *This is overall a well written and motivated manuscript, although heavy on jargon, technical language and acronyms in places, and assuming the reader is an 'expert' in some places without full explanations. The manuscript is likely aimed at a very specific audience, and does little to make it more accessible to those who might be at the periphery of snowpack simulations. That said, it is clearly a substantive work which will be good for those doing snowpack simulations, although they will need to read it several times to fully understand what is being done and how it is being done.*

*I have made a series of suggestions below, in no order of importance, but rather notes as reading through it, and sometimes returning back to different parts of the manuscript. Because no major issues came up, I would suggest that this go through minor changes. Many of the following are stylistic and to improve the structure/content, with an occasional query about meaning and details given.*

**Author Comment**: Thank you very much for taking the time to review our manuscript! We appreciate and value your perspective, particularly on accessibility. We respond to each comment in a point-by-point manner below.

**1.2 Detailed comments**

**1.2.1 Abstract: quantitative summary**

Referee Comment: *1. [Abstract] Consider whether to put more quantitative summaries into the abstract (currently is very narrative).*

**Author Response:** Thank you for the suggestion. In our revised abstract, we include more quantitative summaries. We also used your prompt to tighten up the narrative writing style a bit to keep the word count at bay. The revised abstract reads as follows (see highlighted text for added statements):

> Avalanche warning services increasingly employ snow stratigraphy simulations to improve their current understanding of critical avalanche layers, a key ingredient of dry

slab avalanche hazard. However, a lack of large-scale validation studies has limited the operational value of these simulations for regional avalanche forecasting. To address this knowledge gap, we present methods for meaningful comparisons between regional assessments of avalanche forecasters and distributed snowpack simulations. We applied these methods to operational data sets of ten winter seasons and three forecast regions with different snow climate characteristics in western Canada to quantify the Canadian weather and snowpack model chain's ability to represent persistent critical avalanche layers.

Using a recently developed statistical instability model as well as traditional process-based indices, we found that the overall probability of detecting a known critical layer can reach 75 % when accepting a probability of 40 % that any simulated layer is actually of operational concern in reality (i.e., precision) as well as a false alarm rate of 30 %. Peirce skill scores and F1 scores cap at approximately 50 %. Faceted layers were captured well but also caused most false alarms (probability of detection up to 90 %, precision between 20–40 %, false alarm rate up to 30 %), whereas surface hoar layers, though less common, were mostly of operational concern when modeled (probability of detection up to 80 %, precision between 80–100 %, false alarm rate up to 5 %). Our results also show strong patterns related to forecast regions and elevation bands and reveal more subtle trends with conditional inference trees. Explorations into daily comparisons of layer characteristics generally indicate high variability between simulations and forecaster assessments with correlations rarely exceeding 50 %. We discuss in depth how the presented results can be interpreted in light of the validation data set, which inevitably contains human biases and inconsistencies.

Overall, the simulations provide a valuable starting point for targeted field observations as well as a rich complementary information source that can help alert forecasters about the existence of critical layers and their instability. However, the existing model chain does not seem sufficiently reliable to generate assessments purely based on simulations. We conclude by presenting our vision of a real-time validation suite that can help forecasters develop a better understanding of the simulations' strengths and weaknesses by continuously comparing assessments and simulations.

**1.2.2 Introduction**

Referee Comment: *2. Insert somewhere in the first paragraph of the introduction "the subject of this paper" so it is clear that this will be the subject to be discussed.*

*3. The introduction is strong, well cited, but would benefit (because of its length of five paragraphs, a sentence at the end of the first paragraph stating something like "In the rest of this introduction we will…" to signal to the reader where you are headed.*

**Author Response:** We appreciate your suggestions for enhancing the clarity and structure of our introduction. In response to both of your comments, we have added a sentence to the end of the first paragraph to clearly define the subject of the paper. We believe that the suggested outline would have disrupted the flow and storyline of our introduction, which follows a conventional and widely recognized structure. The inserted sentence reads

> This paper examines the effectiveness of the Canadian operational weather and snowpack model chain to identify critical avalanche layers that are essential for regional-scale avalanche forecasting.

**1.2.3 Outline Section 2**

Referee Comment: *4. After the first sentence of Section 2, tell us how the Section 2 will be organized.*

**Author Response:** Good idea. We added the following sentence.

> This section provides the necessary background information on the study area (Sect. 2.1), the snowpack simulations employed for this study (Sect. 2.2), as well as the human hazard assessments used as validation data set (Sect. 2.3).

**1.2.4 Winter season definition**

Referee Comment: *5. Line 81. Indicate here or elsewhere the normal months for the winter season. In particular, if you state a winter season of 2013, is this from 2012 to 2013 or 2013 to 2014.*

*14. Section 2.3 " of the winter season" Again, please define or give us an idea of how the winter season varies, or if the same months are used, which these are. If it is just the standard definition of November to March, that is fine, but state, and whether or not border line months (or other months) also would become important.*

**Author Response:** Sure. We added the following statements:

> Sect. 2: In the manuscript, each winter season is defined to span from December of the previous year through March of the stated year (e.g., winter season 2021/22 will be referred to as 2022).

> Sect. 2.2: The simulations were initialized in September without any snow on the ground.

**1.2.5 Figure 1**

Referee Comment: *6. Figure 1. Put ALP, TL, BTL into figure caption where they appear (alpine [ALP], etc.). For all figure captions, please ensure that you acknowledge source of data explicitly in the figure caption.*

**Author Response:** Done.

Referee Comment: *7. Figure 1. Please include a scale for S2S, GNP, BYK insets. For overall figure map, put the line indicating 100 km slightly lower, so one can really see that it is the 'length' for 100 km. In figure caption, indicate size of grid cell (in addition to where it is mentioned in the text).*

**Author Response:** Done. Added statement to caption: "The grid has a 2.5 km spacing."

**1.2.6 Unit: m asl**

Referee Comment: *8. Line 95 and other locations, m is m asl? If so, be clear.*

**Author Response:** Thanks, we corrected that!

**1.2.7 Study area coverage**

Referee Comment: *9. Line 92. "Overall, we selected 1004 grid points (Fig. 1) covering an area of ******".*

**Author Response:** We included the statement. "covering an area of 6275 km$^2$".

Referee Comment: *10. For elevations within your classes S2S, GNP, BYK, it would be good to know the distribution of the elevation points, and some idea of which way these slopes are facing, along with any prominent wind directions.*

**Author Response:** Absolutely, we will include a table of grid point distributions (Table 1—please note that the table will render in the typical Copernicus style in the actual manuscript). The simulations are all run with flat field conditions and without wind transportation schemes (see next Section, first paragraph, 2.2 Snowpack simulations). Therefore we did not include a discussion of slope aspects or prominent wind directions.
We added the following sentences to Section 2.1:

> Table 1 describes the distribution of model grid points across the forecast regions and elevation bands. Due to the configuration of our snowpack simulations (flat field, no wind transport, see next section, 2.2), we do not discuss slope aspects or prominent wind directions across our study areas.

**1.2.8 Table of acronyms and variables**

Referee Comment: *11. Because of the large number of acronyms used in this paper, I recommend that early on you have a Table of Acronyms (Table 1) to make it easier for the reader in what is a fairly 'dense' paper.*

*21. Table of variables (could be combined with table of acronyms). There are a lot of variables—consider having a table of acronyms and variables introduced early on, defining each variable, name, units, etc.*

**Author Response:** Sure. We added the tables to an appendix (please note that the tables will render in the typical Copernicus style in the actual manuscript), but reference the tables early in the manuscript.

> A comprehensive table of acronyms (Table 2) and table of variables (Table 3) can be found in Appendix B.

> [...]

> This appendix provides a table of acronyms (Table 2) and a table of variables (Table 3) for abbreviations and symbols used throughout the manuscript.

| | TOTAL | alpine (ALP) | treeline (TL) | below treeline (BTL) |
|---|---|---|---|---|
| Sea-to-sky (S2S) | 476 | 48 (10%) | 54 (11%) | 374 (79%) |
| Glacier National Park (GNP) | 233 | 30 (30%) | 102 (44%) | 101 (43%) |
| Banff–Yoho–Kooteney (BYK) | 295 | 7 (2%) | 163 (55%) | 125 (42%) |

Table 1: Number and percentage of model grid points in each region and elevation band.

| Acronym | Description |
|---|---|
| ALP | Alpine elevation band |
| BTL | Below treeline elevation band |
| BYK | Banff-Yoho-Kooteney National Park |
| CTree | Conditional inference tree |
| DF | Decomposing and fragmented particles |
| DH | Depth hoar |
| FC | Faceted crystals |
| FCxr | rounding facteted particles |
| $F_1$ | $F_1$ skill score |
| FN | False negative result |
| FP | False positive result |
| GNP | Glacier National Park |
| HRDPS | High resolution deterministic prediction system, a numerical weather prediction model |
| IFrc | Rain crust |
| IFsc | Sun crust |
| MF | Melt forms |
| MFcr | Melt–freeze crust |
| nWKL | Number of weak layers |
| PSS | Peirce skill score |
| PP | Precipitation particles |
| RG | Rounded grains |
| ROC | Receiver operating characteristics curve |
| RTA | Relative threshold sum approach (snow stability index) |
| S2S | Sea-to-Sky avalanche forecast region |
| SH | Surface hoar |
| SK38 | Skier stability index |
| TL | Treeline elevation band |
| TN | True negative result |
| TP | True positive result |

Table 2: Descriptions of acronyms used throughout the manuscript.

| Variable | Name (unit) | Description |
|---|---|---|
| $\Delta_{\text{duration}}$ | Difference (days) | Agreement indicator of layer instability |
| $< \frac{\text{density}}{\text{grain size}} >_{\text{slab}}$ | slab cohesion (kg m$^{-3}$ mm$^{-1}$) | Average ratio of density over grain size of the slab |
| $< \text{HN24} >_{75}$ | Solid precipitation (m) | 75th percentile of 24 hour new snow amounts |
| $< \text{HN72} >_{25}$ | Solid precipitation (m) | 25th percentile of 72 hour new snow amounts |
| $\Lambda_{\text{onset}}$ | Lag (days) | Agreement indicator of layer instability |
| $\Lambda_{\text{turn-off}}$ | Lag (days) | Agreement indicator of layer instability |
| $p_{\text{unstable}}$ | Probability of layer instability (1) | A snow stability index |
| $< \text{RAIN24} >_{75}$ | Liquid precipitation (m) | 75th percentile of 24 hour rain amounts |
| $< \text{RAIN72} >_{25}$ | Liquid precipitation (m) | 25th percentile of 72 hour rain amounts |
| $r_c$ | Critical crack length (m) | A mechanical snow layer property |
| $\rho_\Psi$ | Spearman rank correlation (1) | Agreement indicator of layer instability |

Table 3: Names, units, and description of variables used throughout the manuscript.

**1.2.9 Typo**

Referee Comment: *12. Line 99, Lehning et al., 2002a, b (not b, a)*

**Author Response:** Done.

**1.2.10 Spell out acronyms**

Referee Comment: *13. General: first time important acronyms are given, spell out, e.g., HRDPS (High Resolution Deterministic Prediction System)*

**Author Response:** Done.

**1.2.11 Include extra figure of data set**

Referee Comment: *15. Section 2 for data, I would have liked to have seen perhaps 1-2 other figures (e.g., photos, bulletins, maps) representing the real data that was used. Not a strong requirement on this, but it would have been helpful to bring this back to reality of what is being modelled for the reader.*

**Author Response:** That is a good idea, thank you! We do not have a suitable and pleasing visual yet that displays the human hazard assessment, but are trying to get a meaningful screenshot from Avalanche Canada's forecasting dashboard. We also compiled a figure that shows a snowpack simulation at a single grid point to characterize the simulations in the data section. We aim for adding these figures to the revised manuscript but can not show them here yet.

**1.2.12 Figure 2**

Referee Comment: *16. Figure 2. These colours do not work in the PDF downloaded, and are very difficult to read. For example, white on bright green or white on bright blue, are not recommended. Font size getting too small. Figure caption, define all acronyms, and tell us what the different colours mean. This could be overall a stronger flowchart and figure caption as currently it would need the author next to the reader to explain what they are seeing.*

**Author Response:** Thank you very much for the heads up on this! We changed the text color to black, which should be a lot easier to read. We also increased the font size of the smallest annotation elements. And we expanded the caption (see Fig. 1) to (1) explain the meaning of different colors, (2) explain the confusion matrix in more detail, and (3) tell the reader that all visualized processes are explained in detail in Sections 3.1.1–3.1.5.

**1.2.13 Figure 3**

Referee Comment: *17. Figure 3. Similar to figure 2 in terms of colours used (hard to distinguish all of these). Perhaps use https://colorbrewer2.org/ to help you pick your colour palettes. Rather than refer us to Sect. 2.2. for colours, refer us to a table or put them actually in the caption. Mostly well explained in terms of the figure caption, except I did not follow the dashed horizontal line, and why the vertical grey line on the left of the formation line goes 'before' the arrow. I did not get 'time' here—the text states that the burial window is four days, but the solid line for the burial window is 3 days and the dash line 2 days. Can time be made clearer in text caption and the figure, that each box horizontally represents one day (I think)?*

**Author Response:**

[Figure]

Figure 1: (a) A flowchart for assembling the four cells of the confusion matrix from the human and modeled data sets. The human data set is displayed in green boxes, snowpack simulations in turquoise, data points informed by both data sets in dark blue. (b) The resulting confusion matrix categorizes all critical layers into two dimensions: whether they are of human concern and whether they are identified by the simulation. All processes visualized by this figure are explained in detail in Sect. 3.1.1–3.1.5.

- The color palette used for grain types originates from research on designing the optimal color palette for displaying snow grain types to avalanche forecasters (Horton et al. 2020). We therefore keep the current colors.

- We reworded the reference to the acronym definitons: "Colors refer to snow grain types; acronyms for these grain types are defined in Sect. 2.2 and Table A1."

- Referee 1 already made a similar comment on the caption, so the caption was already revised to better explain the horizontal lines. After your comment, we specifically revised the sentence about the dashed horizontal line and also removed the small arrow, which seems to have rather caused confusion than helped comprehension.

- Thanks for examining this figure so closely and spotting an error on our end: The burial window is indeed limited to 5 days (like displayed in the Figure) and not 4 days (like described in the text). We changed the text accordingly.

- We included a label in the Figure that highlights that each column refers to one day and we also added it to the caption. The revised caption reads:

  An illustration of the search windows for layers of human concern around the human date tag. The panel zooms in on the near-surface layers of a time series of a simulated snow profile (daily time step). The extent of the formation and burial windows are highlighted by horizontal lines, where the solid lines indicate the extent of each window based on the timing of the storms. The black boxes highlight all layers that either formed within the formation window or got buried within the burial window. To better illustrate the effect that different lengths of (potentially fixed) time windows have on the selection of layers, the gray hatched areas highlight layers that would result from time windows extended by the dashed horizontal lines. Colors refer to snow grain types; acronyms for these grain types are defined in Sect. 2.2 and Table A1.

**1.2.14 Figure 4**

Referee Comment: *18. Figure 4. Similar comments as above, but I was unclear about the dash vertical vs. the solid vertical line and dark grey vs. light grey in 'a'. What is the dark horizontal line in 'a'? For part 'd', can you move the 'of concern' over a bit to the 'yes' and 'no' so it is not confused as being a secondary axis for 'c'? Please make clearer the time axis, that this is 2019 to 2020 (I think). This can be signalled in the axis and in the figure caption. When I pair the figure with the text, there is a lot that feels left out in explanation (either in figure caption or in the text) and again, it almost needs the authors to be with the readers to explain each aspect. For the violin plot in 'b', do you not need to have 0.0 to 1.0 on y-axis, and define what is meant by the white dot, and the black bars (there are MANY ways of doing violin plots, you cannot assume a give way is being shown here that everyone will automatically understand).*

**Author Response:**

- We made the explanation of the vertical lines (date tags) and the horizontal line in (a) clearer in both text and caption. Please consider that this paragraph is an applied example of the methodology. So a reader can not expect to understand this paragraph without having read the methods before.

- We moved the label 'of concern' over to the right as requested.

- We added labels to make the years 2018 and 2019 obvious in the figure and also state the winter season in the caption.

- We added an explanation of the violin plot to the caption.

- The y axis is shared between panels (a) and (b), so we omitted the labeling in (b) to save some horizontal space and instead show the figure a bit larger.

- The revised paragraph and the figure:

  To summarize our methodology, we illustrate the concepts applied so far with the 2019 winter season in GNP at TL (Fig. 2). The time series of the average profile (Herla et al. 2022) for the region, shown in the bottom left panel, provides context for the date tags that mark the beginning of important snowfall periods (Sect. 2.3, 3.1.2, 3.1.3). Date tags that were reported by forecasters are visualized by solid red vertical lines while the additional model-derived date tags are indicated by dashed gray vertical lines. For each regional layer represented by its date tag, a bar in the bar chart of the top left panel represents the maximum daily proportion unstable. Using 50 % as our threshold criterion for this example, each bar is colored according to its corresponding cell in the confusion matrix in the bottom right panel. The violin plots shown in the top right panel present the distributions of the maximum daily proportion unstable in more detail, while also adhering to the same colors. In this particular case, all six layers of concern (represented by the red bars, red vertical lines, red violin, and dark red cell of the confusion matrix) were well captured, and all the other simulated layers (represented by all gray features) generally had lower proportions of unstable grid points. However, there are five layers that were considered critical by the model using the 50 % threshold but not the human forecasters.

**1.2.15 Page number in citation**

Referee Comment: *19. [Minor] Line 273. Why are you putting in a page number (unless this is a direct quote, in which case you should have " ").*

**Author Response:** It is not a direct quote. However, here we cite a book that is several hundred pages long. So putting the page number to the Section of the exact topic we refer to is only to make the interested reader's life easier.

**1.2.16 Comma after equation**

Referee Comment: *20. Equation 2: What is FN'? [You define FN, but not FN']. Ah, never mind, I see now that it is a comma, but looks like a '*

**Author Response:** This is a comma indeed. We removed it to not have anyone else fall into this trap again. Thanks for pointing it out.

**1.2.17 Discussion section 5.1**

Referee Comment: *22. Section 5.1 is a substantive discussion on insights.*

*Can these be better broken out rather than having almost three pages of narrative text, so as to make this easier for higher level reading.*

*Can this be brought back a bit more to the broader literature (two citations seems really few for bringing this back to the wider community and what has been done).*

[Figure]

Figure 2: An illustration of the concepts applied to a case example of the 2019 season in Glacier National Park (GNP) at treeline elevation (TL). (a) Maximum daily proportion unstable for each regional layer (red: of human concern, gray: not of human concern) and an exemplary threshold criterion of 50 % (black horizontal line; shades of gray bars depend on the threshold criterion: dark gray corresponds to 'not modeled', light gray corresponds to 'modeled') and (b) the resulting distributions (colors are shared with panels a and d; boxplots within violins represent the median and interquartile range). (c) The average profile for the region and season (only shown to improve context; black shading highlights times and layers of modeled instability) with the human and model-derived date tags (red and gray vertical lines, respectively). (d) The confusion matrix evaluated for this specific example. Acronyms of snow grain types listed in the legend are defined in Sect. 2.2 and Table A1.

**Author Response:** In the revised manuscript, we added three subsections to brake out Section 5.1:

- A tangible interpretation of overall model performance

- Performance variations by grain types, elevation bands, and forecast regions

- Detailed comparisons of human–modeled data set

You are correct in that we do not cite many other references in this Section 5.1 (4 citations to be precise). This has the following two reasons. First, the study we present follows a validation approach that has not been carried out before (i.e., application-specific validation of critical layers on the regional scale against human hazard assessments over many seasons). Therefore there are naturally not many other references that we can meaningfully compare our results against. Due to our validation design and the simulation of snowpack stability there are indeed many interfaces to other studies. We write and cite about all these connection points in the relevant sections of the Discussion (5.2: six citations, 5.3: three citations, 5.4: seven citations), which brings us to the second reason. This Section 5.1 is specifically meant to elicit the take home points of this study, made palatable to avalanche forecasters and snowpack modelers working on applied solutions, and focusing on the Canadian context. On this intersection, there are not many other publications to bring this back to. Overall we think the Discussion as a whole finds a good balance of tying our study to the existing research.

**References**

Herla, F., Haegeli, P., and Mair, P.: A data exploration tool for averaging and accessing large data sets of snow stratigraphy profiles useful for avalanche forecasting, The Cryosphere, 16, 3149–3162, https://doi.org/10.5194/tc-16-3149-2022, 2022.

Horton, S., Nowak, S., and Haegeli, P.: Enhancing the operational value of snowpack models with visualization design principles, Nat Hazard Earth Sys, 20, 1557–1572, https://doi.org/10.5194/nhess-20-1557-2020, 2020.